# FROM LOGITS TO HIERARCHIES:
# HIERARCHICAL CLUSTERING MADE SIMPLE

## ABSTRACT

The structure of many real-world datasets is intrinsically hierarchical, making the modeling of such hierarchies a critical objective in both unsupervised and supervised machine learning. Recently, novel approaches for hierarchical clustering with deep architectures have been proposed. In this work, we take a critical perspective on this line of research and demonstrate that many approaches exhibit major limitations when applied to realistic datasets, partly due to their high computational complexity. In particular, we show that a lightweight procedure implemented on top of pre-trained non-hierarchical clustering models outperforms models designed specifically for hierarchical clustering. Our proposed approach is computationally efficient and applicable to any pre-trained clustering model that outputs logits, without requiring any fine-tuning. To highlight the generality of our findings, we illustrate how our method can also be applied in a supervised setup, recovering meaningful hierarchies from a pre-trained ImageNet classifier.

## 1 INTRODUCTION

Modeling hierarchical structures in the data is a long-standing goal in machine learning research (Bengio et al., 2013; Jordan & Mitchell, 2015). In many real-world scenarios, data is inherently organized in hierarchies, such as phylogenetic trees (Linnæus, 1758; Sneath & Sokal, 1962; Penny, 2004), tumor subclasses (Sørlie et al., 2001) and social networks Ravasz & Barabási (2003); Crockett et al. (2017). In unsupervised learning, hierarchical clustering can provide more accurate insights than flat (i.e. non-hierarchical) clustering methods by introducing multiple levels of granularity and alleviating the need for a fixed number of clusters specified a priori (Bertsimas et al., 2021; Chami et al., 2020). This aids scientific understanding and interpretability by providing a more informative representation (Lipton, 2018; Marcinkevičs & Vogt, 2020). The benefits of modeling a hierarchy in the data extend to supervised scenarios. For example, interpretable methods based on decision trees (Breiman, 2001; Tanno et al., 2019) hierarchically partition the data so that points in each split are linearly separable into classes. More recent work leverages hierarchies in the data to improve supervised methods (Bertinetto et al., 2020; Goren et al., 2024; Karthik et al., 2021) or for self-supervision (Long & van Noord, 2023).

Among classic algorithms for hierarchical clustering, agglomerative methods have been the most widely adopted. These methods compute pairwise distances between data points, often in a lower-dimensional representation space. Starting from the instance level, a hierarchy is then built based on the pairwise distances by recursive agglomeration of similar points or clusters together in a bottom-up fashion (Murtagh & Contreras, 2011). More recently, a revived interest in hierarchical clustering has sparked novel approaches using deep architectures (Mautz et al., 2020; Goyal et al., 2017; Shin et al., 2019; Vikram et al., 2019; Manduchi et al., 2023). However, despite their promising methodological contributions, recent approaches require specialized architectures and complex training schemes. Consequently, they cannot be applied to large-scale datasets due to their expensive computational requirements. Moreover, we find that they often exhibit a lower performance at the leaf level compared to non-hierarchical models.

In this work, we take a critical perspective on recent research on hierarchical clustering and offer a simple alternative. Instead of designing specialized hierarchical clustering models, we develop a lightweight method for hierarchical clustering given a pre-trained flat model. In particular, we show that a lightweight algorithm implemented on top of (non-hierarchical) pre-trained models markedly

outperforms specialized models for hierarchical clustering. Notably, our algorithm, which we name Logits to Hierarchies (L2H), *only uses logits* and *requires no fine-tuning* of the pre-trained model. Hence, it generally applies to black-box models even without access to internal representations (e.g., API calls to proprietary models) and bypasses the costly computation of pairwise distances between data points. Moreover, it also applies to supervised models, for which the inferred hierarchy of classes can aid model interpretability, e.g., for discovering potential biases such as spurious correlations between classes.

In summary, we make the following key contributions:

- In Section 3, we propose a simple algorithm for hierarchical clustering that transforms the logits from a pre-trained model into a hierarchical structure of classes. Our method markedly outperforms specialized hierarchical models and has low computational requirements. With logits as input, it computes a hierarchical clustering on ImageNet-sized datasets in a few minutes on a single CPU core.

- In Section 4.1, our experiments reveal significant limitations of recently proposed methods for hierarchical clustering, highlighting their weaknesses on large-scale datasets and subpar performance at the leaf level compared to non-hierarchical approaches.

- In Section 4.2, we provide a case study on ImageNet to demonstrate how our method applies to supervised models, showing how the inferred hierarchy of classes recovers parts of the WordNet hierarchy and helps discover potential biases of the pre-trained model and ambiguities in existing categorizations.

## 2 RELATED WORK

Hierarchical clustering aims to learn clusters of data points that are organized in a hierarchical structure. The methods used can be broadly categorized into agglomerative and divisive approaches (Nielsen, 2016). The former tackles the problem with a bottom-up approach and iteratively agglomerates clusters into larger ones until a full hierarchy is built in the form of a dendrogram, starting with each datapoint being a separate cluster (Murtagh & Contreras, 2011). The similarity of data points is measured according to a distance function, which for high-dimensional data is often defined on a lower-dimensional representation space. Multiple linkage methods have been proposed to compute the distance between clusters of data points formed at a given step of the algorithm (Sneath, 1957; Ward, 1963). As examples, *single*, *average*, and *complete* linkage characterize the distance between two clusters as the minimum, average, and maximum distance between their data points, respectively. Since these algorithms can be costly, particularly in high-dimensional spaces, approximate versions have been developed for faster computation (Abboud et al., 2019; Cochez & Mou, 2015). Notably, linkage methods are still widely applied in many domains, for instance, in medical research (Nguyen et al., 2024; Senevirathna et al., 2023; Resende et al., 2023).

On the other hand, divisive algorithms start with all objects belonging to the same cluster and recursively split them into subclusters. While early approaches are mostly based on heuristics, Dasgupta. (2016) proposed the Dasgupta cost: an objective function for evaluating a hierarchical clustering, with a divisive approach to provide an approximately optimal solution. HypHC introduces a continuous relaxation of Dasgupta's discrete optimization problem with provable guarantees via hyperbolic embeddings that better reflect the geometry of trees compared to Euclidean representations Chami et al. (2020); Liu et al. (2019). More recently, research has focused on developing deep learning approaches for hierarchical clustering (Mautz et al., 2020; Goyal et al., 2017; Shin et al., 2019; Vikram et al., 2019; Manduchi et al., 2023). Among these, DeepECT learns a hierarchical clustering on top of the embedding space of a jointly optimized autoencoder (Mautz et al., 2020). Notably, TreeVAE not only learns a hierarchical clustering in the latent space but also provides a generative model that adheres to the learnt hierarchy, thereby enabling sample generation in a structured manner (Manduchi et al., 2023). However, all these approaches have mostly been tested on simple datasets, far from realistic settings. As shown in our experiments, they present relevant limitations when deployed on more challenging datasets, mainly due to their high computational complexity.

Finally, the benefits of modeling a hierarchy in the data are not restricted to the unsupervised setup. In particular, a recent line of research focuses on leveraging a tree structure in the classes to assess and reduce the severity of misclassification of supervised models (Karthik et al., 2021). This can lead

to safer models in cost-sensitive classification scenarios (Bertinetto et al., 2020) and allow a classifier to predict at different levels of the hierarchy depending on the required confidence (Goren et al., 2024). The visualization of hierarchies also provides global explanations of a model's functionality, thereby improving a user's understanding of the model behavior and fostering trust (Chakraborty et al., 2017; Lipton, 2018).

## 3 METHOD

In this work, we take a critical perspective on a recent line of research on hierarchical clustering: as an alternative to designing ad-hoc complex approaches, we focus on adapting pre-trained flat models to output a hierarchy with minimal overhead. To this end, we introduce a lightweight algorithm to leverage the information contained in the logits of a pre-trained flat clustering model to output a hierarchy of clusters. In the following, we describe the proposed procedure and also provide a graphical illustration as well as detailed pseudocode.

Let $\mathcal{D} = \{\boldsymbol{x}_1, \ldots, \boldsymbol{x}_N\}$ be a dataset consisting of $N$ data points and $f_\theta$ be a non-hierarchical model trained to partition $\mathcal{D}$ into $K$ clusters. We assume that $f_\theta$ outputs unnormalized logits, i.e. the cluster assignment $k^*$ for a datapoint $\boldsymbol{x}$ is determined by computing $k^* = \arg\max_{k \in \{1,\ldots,K\}} \mathrm{softmax}_k(f_\theta(\boldsymbol{x}))$. We define two functions

$$h_\theta(\boldsymbol{x}) = \underset{k \in \{1,\ldots,K\}}{\arg\max} \; \mathrm{softmax}_k(f_\theta(\boldsymbol{x})) \qquad g_\theta(\boldsymbol{x}) = \max_{k \in \{1,\ldots,K\}} \mathrm{softmax}_k(f_\theta(\boldsymbol{x}))$$

of which $h_\theta$ computes the cluster assignment for a datapoint $\boldsymbol{x}$, while $g_\theta$ computes the predicted probability of the cluster assignment for the datapoint $\boldsymbol{x}$.

A key idea behind our method is a simple yet effective way to determine the relatedness of clusters, or groups of clusters, by iteratively grouping them together to construct a hierarchy. Intuitively, to assess which group of clusters $G'$ is most related to a given group $G$, we propose the following strategy. For data points assigned to clusters in $G$, we determine which group $G'$ would have the majority of these data points reassigned to if clusters in $G$ were not available.[1] Formally, we define the following functions to compute cluster assignments and corresponding predicted probabilities, restricting only to a subset of the total set of clusters.

We start with a masked version of the softmax function

$$\mathrm{m\_softmax}_k(\boldsymbol{v}; G) = \begin{cases} \frac{\exp(v_i)}{\sum_{j \in \{1,\ldots,K\} \setminus G} \exp(v_j)} & \text{if } i \notin G \\ 0 & \text{if } i \in G \end{cases}$$

given a $K$-dimensional vector $\boldsymbol{v}$ and a set $G \subset \{1, \ldots, K\}$. This function restricts the softmax operation to the elements of $\boldsymbol{v}$ at indexes in $\{1, \ldots, K\} \setminus G$. Next, we define functions

$$h_\theta^m(\boldsymbol{x}; G) = \underset{k \in \{1,\ldots,K\}}{\arg\max} \; \mathrm{m\_softmax}_k(f_\theta(\boldsymbol{x}); G) \qquad g_\theta^m(\boldsymbol{x}; G) = \max_{k \in \{1,\ldots,K\}} \mathrm{m\_softmax}_k(f_\theta(\boldsymbol{x}); G)$$

Note that the $h_\theta^m$ and $g_\theta^m$ functions correspond to $h_\theta$ and $g_\theta$, except restricting the choice of viable clusters to $\{1, \ldots, K\} \setminus G$. In particular, the $h_\theta^m$ function computes the cluster assignment for a datapoint $\boldsymbol{x}$ restricting to clusters in $\{1, \ldots, K\} \setminus G$, and $g_\theta^m$ the corresponding predicted probability. Lastly, we define

$$\mathcal{D}^c := \{\boldsymbol{x} \in \mathcal{D} \mid h_\theta(\boldsymbol{x}) = c\}$$

i.e. the subset of data points assigned to a given cluster $c \in \{1, \ldots, K\}$. Similarly, we denote as $\mathcal{D}^G = \cup_{c \in G} D^c$ the subset of data points assigned to a group of clusters $G \subset \{1, \ldots, K\}$.

---

[1]This passage is primarily for intuition and not strictly accurate. To be precise, we look not at reassignments but at predicted probabilities of reassignments (see Equation (2)).

---

**Algorithm 1** Logits to Hierarchies (L2H)

---

Given aggregation function $\Lambda$, and functions $g_\theta, g_\theta^m$ defined as above for pre-trained $K$-clustering model $f_\theta$.

---

**Input:** Dataset $\mathcal{D}$.
**Output:** Hierarchy H.
Initialize groups $\mathcal{G} = \{G_1, \ldots, G_K\}$ where $G_k = \{k\}$        # Groups initialized as single clusters
H = []        # Hierarchy initialized as empty list
**for** step $t$ **from** 1 **to** $K - 1$ **do**
     **for** group $G$ **in** $\mathcal{G}$ **do**
         Compute $s(G) := \bigwedge_{\boldsymbol{x} \in \mathcal{D}^c, c \in G} g_\theta(\boldsymbol{x})$        # Compute group scores as in Eqn. 1
     **end for**
     Take $G^\star \in \text{argmin}_{G \in \mathcal{G}} s(G)$        # Select group with lowest score for merging
     **for** cluster $c$ **in** $\{1, \ldots, K\} \setminus G^*$ **do**
         Compute $rp(c) := \sum_{\substack{\boldsymbol{x} \in \mathcal{D}^{G^*} \\ h_\theta^m(\boldsymbol{x}; G^*) = c}} g_\theta^m(\boldsymbol{x}; G^*)$        # Compute total predicted probability
                                                             # per cluster as in Eqn. 2
     **end for**
     Take $G^\dagger = \text{argmax}_{G \in \mathcal{G} \setminus \{G^*\}} \frac{1}{|G|} \sum_{c \in G} rp(c)$      # Select $G^\dagger$, most related group to $G^*$, for merging
     Update $\mathcal{G}$ by merging groups $G^*$ and $G^\dagger$        # Update groups
     Update H by adding that $G^*$ and $G^\dagger$ are merged at step $t$        # Update hierarchy
**end for**

---

We describe our proposed method in Algorithm 1. At the start of the procedure, $K$ groups are initialized as single clusters. [2] At each iteration, two groups are merged into a single group, constructing a tree of clusters up to the root in $K - 1$ iterations. Each iteration can be split in two stages. In the first stage, a score is computed for each group. To compute the score for a given group $G$ we aggregate the predicted probabilities for the data points assigned to clusters contained in $G$ as

$$s(G) = \bigwedge_{\substack{\boldsymbol{x} \in \mathcal{D}^c \\ c \in G}} g_\theta(\boldsymbol{x}) \tag{1}$$

where $\Lambda$ is a chosen aggregation function (e.g. sum function). Then, the lowest scored group $G^*$ is selected for merging at this iteration, which concludes the first stage.

In the second stage, we search for the group $G^\dagger$ that is most related to $G^*$ to perform the merging. To do so, as mentioned above, we look at the subset of data points assigned to clusters in $G^*$: for these data points we recompute cluster assignments and predicted probabilities, this time restricting to clusters not contained in $G^*$. More formally, the total reassigned predicted probability to each cluster not contained in $G^*$ is computed as

$$rp(c) := \sum_{\substack{\boldsymbol{x} \in \mathcal{D}^{G^*} \\ h_\theta^m(\boldsymbol{x}; G^*) = c}} g_\theta^m(\boldsymbol{x}; G^*) \qquad \forall c \in \{1, .., K\} \setminus G^* \tag{2}$$

Note that this quantity can be interpreted as a measure of relatedness between each cluster $c \in \{1, .., K\} \setminus G^*$, and the group of clusters $G^*$. The most related group to $G^*$ is finally selected as $G^\dagger \in \text{argmax}_{G \in \mathcal{G} \setminus \{G^*\}} \frac{1}{|G|} \sum_{c \in G} rp(c)$, i.e. by averaging the total reassigned predicted probability across clusters in each group and selecting the group with the highest average. Given that cluster assignments and corresponding predicted probabilities can be computed via simple operations on the logits, the whole procedure can be executed with as input only the logits for the dataset $\mathcal{D}$ outputted from a pre-trained model $f_\theta$. To show this, we report an example Python implementation in Appendix A. Additionally Figure 1 provides an illustration to exemplify the proposed grouping strategy.

---

[2] Note that here *cluster* is used to refer to a single cluster found by the pre-trained model, while *group* refers to a set of clusters that are grouped together at a given step of the algorithm.

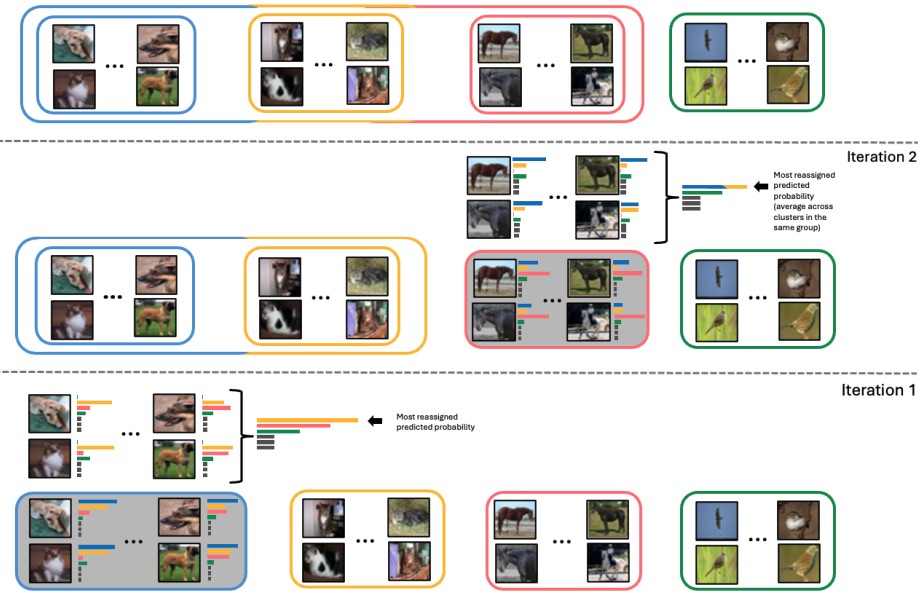

Figure 1: Illustration of the L2H algorithm. The four depicted clusters represent *dogs* in blue, *cats* in yellow, *horses* in red, *birds* in green respectively. In the first iteration (bottom), where groups correspond to single clusters, the dog cluster is selected for merging (shaded in grey). When recomputing predicted probabilities for samples in the dogs cluster, restricting to the remaining clusters, the cluster of cats has the highest predicted probability of reassignment. Note how, after merging, these two clusters are considered as a single group in the next iteration (top).

## 4 EXPERIMENTS

In this section we showcase the experimental results obtained with our proposed approach. In the first part, we focus on the task of hierarchical clustering. We demonstrate that existing specialized models for hierarchical clustering present major limitations when applied in realistic settings. In contrast, our proposed approach achieves convincing results in challenging vision datasets, markedly outperforming alternative methods. In the second part of this section, we present a case study where we discuss the application of the L2H algorithm on top of a pre-trained ImageNet classifier, demonstrating its value for model interpretability and the discovery of spurious correlations. Additional experimental details on datasets, implementations and metrics can be found in Appendix B.

### 4.1 HIERARCHICAL CLUSTERING

In this section, we compare the performance of our proposed method for hierarchical clustering with recent specialized approaches on three challenging vision datasets: namely the CIFAR-10, CIFAR-100 (Lake et al., 2015) and Food-101 (Bossard et al., 2014) datasets. We report our results in Table 1. For each dataset, we implement our algorithm on top of two pre-trained flat clustering models, namely TURTLE (Gadetsky et al., 2024) and TEMI (Adaloglou et al., 2023). These are two state-of-the-art clustering methods (see Appendix B.2 for more details), both of which are *not* designed to produce a hierarchy of clusters. In our evaluation, we report both metrics to evaluate models at the flat level and metrics to evaluate the quality of the produced hierarchy. For comparing models at the flat level, we report Normalized Mutual Information (NMI), Adjusted Random Index (ARI), Accuracy and Leaf Purity (LP). To assess the quality of the hierarchical clustering, we report two metrics: Dendrogram Purity (DP) and Least Hierarchical Distance (LHD). The former was introduced in Kobren et al. (2017) and extends the notion of purity, normally evaluated at the leaf level, to assess the quality of a tree clustering: higher purity corresponds to higher quality of the hierarchy. Note that this metric was recently adopted in Manduchi et al. (2023) to benchmark deep hierarchical clustering models. Least Hierarchical Distance, on the other hand, measures the average

minimal log-distance in the hierarchy between any pair of data points that have the same true label but different cluster assignments. A better hierarchy corresponds to a lower LHD. More details about our metrics can be found in the Appendix B.3.

The results in Table 1 uncover the aforementioned limitations of recent deep learning methods (DeepECT, TreeVAE). In particular, these models do not scale well in terms of the depth of the hierarchy, thereby producing overly shallow hierarchies for datasets with a large number of classes (CIFAR-100, Food101). We find this to be linked to their high computational complexity, which we discuss later in Table 2. For instance, TreeVAE learns a hierarchical generative model with leaf-specific decoders: this choice helps its performance in a generative scenario but impacts its scalability to large-scale datasets. Importantly, the comparison in terms of flat clustering metrics highlights that ad-hoc hierarchical models produce clusterings at the leaf level that are much less accurate than those obtained with non-hierarchical models. The results for DP and LHD highlight the shortcomings in these tasks. Note that we notice the presence of artifacts, underlined in Table 1, for the LHD metric, due to overly shallow hierarchies that degenerate to having only two to three leaves.

| | Flat | | | | Hierarchical | | # leaves | Inference on test set |
|---|---|---|---|---|---|---|---|---|
| | NMI ($\uparrow$) | ARI ($\uparrow$) | ACC ($\uparrow$) | LP ($\uparrow$) | DP ($\uparrow$) | LHD ($\downarrow$) | | |
| **CIFAR-10** | | | | | | | | |
| Agglomerative | 0.074 | 0.038 | 0.211 | 0.246 | 0.121 | 0.549 | 10 | ✗ |
| HypHC | 0.019 | 0.009 | 0.134 | 0.359 | 0.104 | 0.569 | 10 | ✗ |
| DeepECT | 0.006 | 0.002 | 0.110 | 0.110 | 0.101 | 0.369 | 2-3 | ✓ |
| TreeVAE | 0.414 | 0.313 | 0.497 | 0.523 | 0.341 | 0.410 | 10 | ✓ |
| L2H-TEMI | 0.901 | 0.906 | 0.956 | 0.958 | 0.902 | 0.348 | 10 | ✓ |
| L2H-Turtle | **0.985** | **0.989** | **0.995** | **0.995** | **0.988** | **0.277** | 10 | ✓ |
| **CIFAR-100** | | | | | | | | |
| Agglomerative | 0.223 | 0.020 | 0.090 | 0.131 | 0.019 | 0.428 | 100 | ✗ |
| HypHC | 0.072 | 0.004 | 0.031 | 0.560 | 0.011 | 0.499 | 100 | ✗ |
| DeepECT | 0.016 | 0.005 | 0.070 | 0.070 | 0.052 | 0.121 | 2-3 | ✓ |
| TreeVAE | 0.199 | 0.098 | 0.228 | 0.242 | 0.103 | 0.484 | 20 | ✓ |
| L2H-TEMI | 0.778 | 0.565 | 0.682 | 0.698 | 0.502 | 0.298 | 100 | ✓ |
| L2H-Turtle | **0.917** | **0.831** | **0.896** | **0.896** | **0.803** | **0.235** | 100 | ✓ |
| **Food-101** | | | | | | | | |
| Agglomerative | 0.082 | 0.004 | 0.039 | 0.045 | 0.011 | 0.438 | 101 | ✗ |
| HypHC | 0.035 | 0.002 | 0.022 | 0.630 | 0.011 | 0.573 | 101 | ✗ |
| DeepECT | 0.003 | 0.000 | 0.011 | 0.011 | 0.010 | 0.333 | 2-3 | ✓ |
| TreeVAE | 0.114 | 0.017 | 0.057 | 0.058 | 0.016 | 0.483 | 20 | ✓ |
| L2H-TEMI | **0.917** | **0.841** | **0.904** | **0.881** | **0.801** | **0.270** | 101 | ✓ |
| L2H-Turtle | 0.894 | 0.800 | 0.876 | 0.843 | 0.758 | 0.297 | 101 | ✓ |

Table 1: Quantitative comparison of hierarchical clustering performance on three datasets (CIFAR-10, CIFAR-100, Food-101). We report as a baseline Agglomerative clustering, deep hierarchical specialized models (DeepECT, TreeVAE), and our L2H method applied on top of two state-of-the-art flat models (TEMI, TURTLE). We also indicate the number of leaves in the hiearchy modelled by each approach, and whether a given method can perform inference on a hold-out test set. We bold best results for each metric and underline results that are artifacts of degenerate solutions with shallow hierarchies. Results are averaged over five runs.

In contrast to alternative approaches, our proposed algorithm recovers high-quality hierarchies for all three datasets, when implemented on top of both TEMI and TURTLE models. Looking at both hierarchical metrics, our method markedly outperforms other approaches, with a consistent margin over costly deep learning specialized methods. These findings demonstrate that L2H can leverage the information embedded in the logits of a pre-trained flat clustering model to model an accurate hierarchy of the clusters, outperforming ad-hoc hierarchical models, even without having access

to internal representations. Note as well that our method does not require any fine-tuning of the pre-trained model and, by construction, retains its clustering performance at the leaf level, which matches state-of-the-art in our results. Finally, the efficacy of our proposed procedure is not hindered by the presence of a large number of the classes in the dataset, as we witness for other methods. In particular, in Appendix C we show that our method can achieve remarkable hierarchical clustering results on datasets as large as ImageNet1K (Deng et al., 2009).

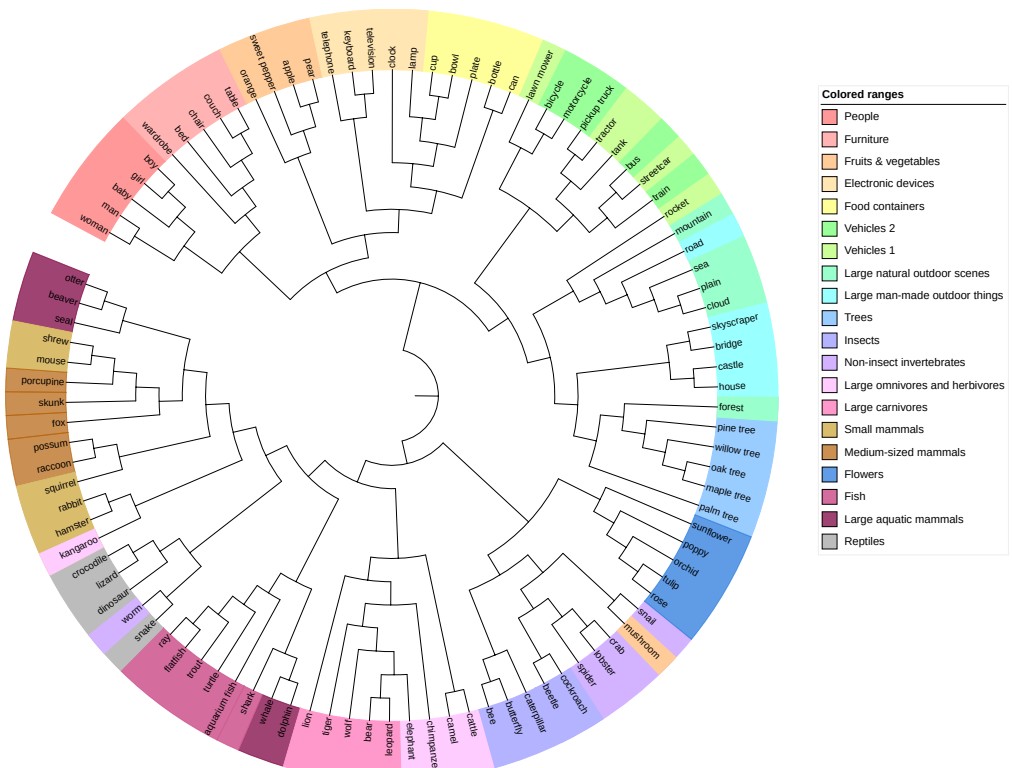

Figure 2: Visualization of the hierarchical clustering produced by L2H-TURTLE on the CIFAR-100 dataset. The inferred hierarchy is represented as a circular tree. On the lowest level, the leaves are annotated by reporting the most frequent label for the samples in each leaf. Leaves are color-colored according to the 20 superclasses in the dataset.

In practice, hierarchical clustering results are often used as a visualization tool, and to analyze the structure of a dataset at different levels of granularity. Hence, to evaluate our proposed approach we visualize and inspect the hierarchy obtained with L2H-TURTLE on the CIFAR-100 dataset in Figure 2. Note that given the absence of leaf labels, we associate a class label to each leaf by looking at the most frequent label among the data points in the given leaf. While an off-the-shelf ground-truth hierarchy is not available for the CIFAR-100 dataset, the authors organize the 100 classes in 20 superclasses. Hence, we color-code the inferred leaf labels in the hierarchy by superclasses and check if the hierarchical clustering recovers this global structure. Notably, the global structure of the superclasses is largely reflected in the visualized hierarchy. Most interesting is that the outliers, for which the color does not coincide with the neighboring leaves, still reflect meaningful semantic associations. For instance, *whale* and *dolphin*—despite being aquatic mammals—are grouped with fish species. However, this is not surprising, given their adaptation exclusively to aquatic environments and the presence of similar traits to fishes, like streamlined bodies. On the contrary, mammals such as *otter*, *beaver*, and *seal*, which are only semi-aquatic, are grouped with other small to medium-sized terrestrial mammals, emphasizing size and communal characteristics like the presence of limbs and fur. Another example is the characterization of *worm* and *snake* alongside in the hierarchy. Although snakes are reptiles, their elongated, limbless bodies visually resemble those of non-insect invertebrates like worms. This showcased analysis confirms the efficacy of our method in recovering a tree structure that follows meaningful semantic associations. The results indicate

that our method produces hierarchies that enable detailed exploration of the structure in the data at varying levels of granularity. Inspecting the hierarchy gives valuable insights for interpretability, revealing underlying associations by the model.

Finally, we end this section with a comparison in terms of the computational cost of our method compared with alternative models, and in particular with specialized deep learning approaches for hierarchical clustering. As the results in Table 2 show, our proposed L2H algorithm is extremely lightweight. For completeness, we also measure the overall runtime to perform hierarchical clustering with our method, including the runtime to train the TURTLE model, as an example model used to compute logits as input. Our approach allows to perform hierarchical clustering extremely efficiently even on large-scale datasets such as ImageNet-1K, with a total training time of a few minutes. Note that, due to the combined efficiency of our method and state-of-the-art flat clustering models, the overall runtime scales seamlessly with dataset size and number of leaves in the hierarchy. Conversely, alternative hierarchical deep learning approaches exhibit a significantly higher computational cost. Moreover, a large dataset size and number of classes markedly increase the computational burden.

| | **Dataset** | | | |
| --- | --- | --- | --- | --- |
| | **CIFAR-10** $K = 10$ $N_{\mathrm{tr}} = 50000$ | **CIFAR-100** $K = 100$ $N_{\mathrm{tr}} = 50000$ | **Food-101** $K = 101$ $N_{\mathrm{tr}} = 75750$ | **ImageNet1K** $K = 1000$ $N_{\mathrm{tr}} = 1281167$ |
| L2H | $< 0.01$ | $< 0.01$ | $< 0.01$ | 0.47 |
| Agglomerative | 0.1 | 0.1 | 0.9 | - |
| HypHC | 163.7 | 153.3 | 195.3 | - |
| DeepECT | 24.1 | 26.2 | 67.5 | - |
| TreeVAE | 364.1 | 756.3 | 2293.7 | - |
| L2H-TURTLE | 1.6 | 1.6 | 1.7 | 5.27 |

Table 2: Training time (in minutes) for our proposed method compared to baselines for hierarchical clustering on CIFAR-10, CIFAR-100, and Food-101 datasets. At the top, we report the runtime for the L2H algorithm alone. Below, we report the runtime of the TURTLE model plus our L2H algorithm to produce a hierarchy, compared with the runtime of each baseline model. Results are averaged over three runs.

## 4.2 CASE STUDY: PRE-TRAINED IMAGENET CLASSIFIER

Next, we demonstrate how our method can be used in a supervised setup to produce a hierarchical clustering given the logits of a pre-trained classifier. Specifically, we use the ImageNet-1K dataset (Deng et al., 2009), which comprises over a million images and a thousand distinct classes with an underlying hierarchical structure. We apply L2H on the logits of a pre-trained ImageNet classifier to determine the hierarchy of classes. As a pre-trained classifier, we use the InternImage model (Wang et al., 2022). The resulting hierarchical clustering is visualized in Figure 3.

Figure 3a shows the inferred dendrogram for the thousand ImageNet classes. The colors indicate whether a leaf node corresponds to the superclass "artifact" or "organism", which we determine based on the corresponding WordNet hypernyms of each class. Overall, we observe a distinct separation between the two superclasses in the inferred dendrogram. Figure 3b zooms into a subtree of the inferred dendrogram that comprises different bird species. Specifically, it shows 58 of the 60 classes of birds found in the ImageNet dataset. The leaf nodes are colored by different clades of bird species (based on the WordNet hierarchy), showing that the inferred hierarchy groups together related species. For example, the group "aquatic bird" is almost completely represented in one of the two main branches, which further splits into a separate cluster for "parrots" and another one for "bird of prey". The other main branch of the tree subdivides further into "passerine" and "game bird" forming distinct clusters.

Overall, our results suggest that the inferred hierarchy recovers a significant portion of the global and local hierarchical structure of the ImageNet dataset given the logits of a pre-trained ImageNet classifier trained with non-hierarchical labels. Yet, the inferred dendrogram also reveals interest-

ing outliers. For example, in Figure 3a, there is a distinct subtree for snow-related artifacts (e.g., dogsled, snowmobile, bobsled) within a large branch of the tree that comprises organisms. Further investigation shows that this group of artifacts is merged with arctic animals (e.g., malamute, Siberian husky, Eskimo dog), which reveals an intuitive correlation between classes and highlights potential biases of the pre-trained model. Likewise, in Figure 3b, we see potential outliers such as "peacock" among the group of parrots or "bustard" among game birds. While the WordNet hierarchy classifies "bustard" as a wading bird, hence among aquatic birds, the inferred hierarchy places it among game birds. Interestingly, common definitions of bustards as terrestrial game birds support the inferred categorization.

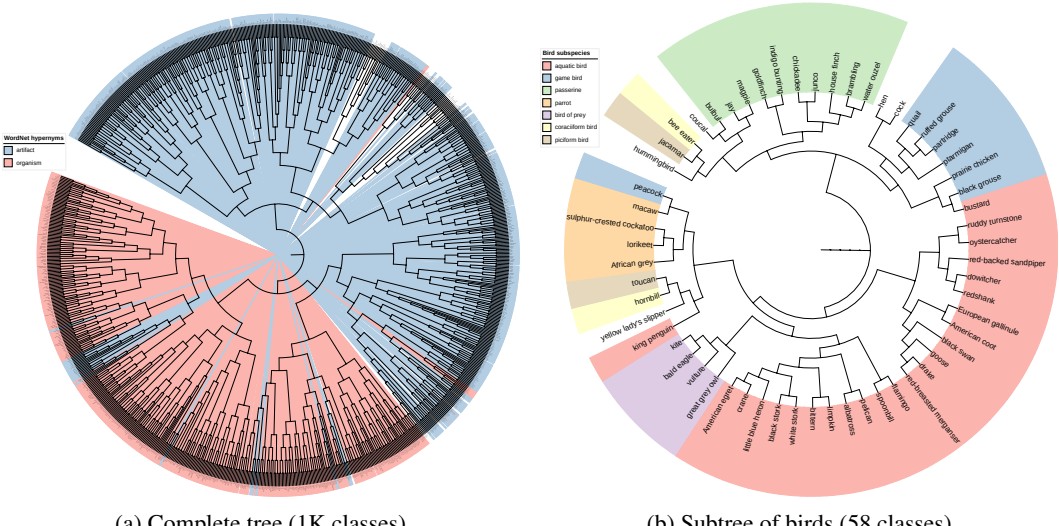

(a) Complete tree (1K classes)     (b) Subtree of birds (58 classes)

Figure 3: Visualization of the hierarchical clustering produced by L2H-TURTLE for the ImageNet-1K dataset. The inferred hierarchy is represented as a circular dendrogram, where the leaf nodes are organized in a circle. Figure 3a shows the complete tree colored by the corresponding WordNet hypernyms "artifact" and "organism", which are the largest two superclasses in the ImageNet dataset. Figure 3b shows the subtree of birds colored by different bird species if they comprise more than one class. The results show that our method recovers a significant portion of the global and local hierarchical structure of the ImageNet dataset.

## 5 CONCLUSION

In this work, we propose a lightweight yet effective procedure for hierarchical clustering based on pre-trained non-hierarchical models. Notably, our solution proves to be markedly more effective and significantly more computationally efficient than alternative methods. Different from existing models for hierarchical clustering, our method can successfully handle large datasets of many classes, taking an important step in deploying hierarchical clustering methods in challenging settings. Moreover, we show that the usefulness of our approach extends to supervised setups, by implementing it on top of a pre-trained classifier to recover a meaningful hierarchy of classes. A case study on ImageNet shows that this approach provides relevant insights for interpretability, and can reveal potential biases of the pre-trained model and ambiguities in existing categorizations.

While we provide extensive results on image datasets, in this work we do not explore other data modalities. However, our procedure is general and may be applied to different data types, which we leave for future work. Hierarchical clustering presents important advantages over non-hierarchical clustering by simultaneously capturing the structure in the data at multiple levels of granularity. However, inspecting the hierarchy is still necessary to extract valuable insights. An important focus for future work is to investigate strategies to partly bypass this process, automatically selecting levels of the hierarchies that provide the most meaningful clustering.

## REPRODUCIBILITY STATEMENT

We share a Python implementation of our L2H algorithm both in Appendix A and in the supplementary material. To ensure the reproducibility of our experimental results, in Appendix B we provide detailed insights on datasets and metrics, as well as on the implementation of our method and the compared baselines.

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

APPENDIX

## A  CODE IMPLEMENTATION OF THE L2H ALGORITHM

In Figure 4, we provide a Python implementation of the L2H algorithm proposed in this work using standard scientific computing libraries (NumPy, SciPy). As stated in Section 3, our algorithm only requires the logits as input. It can be executed on the CPU even for large datasets, e.g., with a runtime of less than a minute for ImageNet. Note that our procedure can be applied to any pre-trained unsupervised model to perform hierarchical clustering. Further, it can also be applied to logits from a supervised model to infer a hierarchy of classes. We store the hierarchy as a list comprised of groups of clusters that are merged iteratively. The aggregation function for computing the score per group is a design choice (as described in Appendix B.2) that can be viewed as a hyperparameter.

```python
import numpy as np
from scipy.special import softmax

def L2H(logits):
    """
    L2H Algorithm.
    Args:
        logits: Logits from model (N x K) where N number of datapoints  in the dataset
                and K is the number of clusters
    Returns:
        steps: Merging steps characterizing the hierarchy

    """
    # Number of clusters is equal to size of last dimension in the logits
    K = logits.shape[-1]
    # Initialize groups of clusters to single clusters
    groups = [(c,) for c in range(K)]
    # Initialize list of steps that characterize hierarchy
    steps = []
    # Given the logits for the whole dataset, compute assignments and predicted probabilities
    softmaxed_logits = softmax(logits, axis=-1)
    assignments = np.argmax(softmaxed_logits, axis=-1)
    pred_probs = np.max(softmaxed_logits, axis=-1)
    for step in range(1, K):
        # Compute score for for each group (which chosen aggregation function)
        score_per_gr = {}
        for group in groups:
            score_per_gr[group] = sum([np.mean(pred_probs[assignments == c]) for c in group])
        # Get the group with the lowest score (lsg), will be merged at this iteration
        lsg = min(score_per_gr, key=score_per_gr.get)
        # Get the logits for datapoints assigned to the lowest score group
        logits_lsg = logits[np.where(np.isin(assignments, lsg))[0]]
        # Reassign datapoints in lsg to only clusters not in lsg,
        # and re-compute predicted probabilities
        msm_logits_lsg = np.zeros_like(logits_lsg)
        cls_not_in_lsg = [c for c in range(K) if c not in lsg]
        cls_in_lsg = [c for c in range(K) if c in lsg]
        msm_logits_lsg[:, cls_not_in_lsg] = softmax(logits_lsg[:, cls_not_in_lsg], axis=-1)
        msm_logits_lsg[:, cls_in_lsg] = 0.
        reassignments = np.argmax(msm_logits_lsg, axis=-1)
        re_pred_probs = np.max(msm_logits_lsg, axis=-1)
        # Compute the total reassigned predicted probability per cluster and average across
        # clusters in each group.Then select the group with the highest average.
        re_pp_per_group = {
            group: np.mean([np.sum(re_pred_probs[reassignments == c]) for c in group]) for
            group in groups if group != lsg
        }
        mtg = max(re_pp_per_group, key=re_pp_per_group.get)
        # Merge `lsg` with `mtg` and update `groups`.
        groups = [gr for gr in groups if gr not in [lsg, mtg]] + [lsg + mtg]
        # Add merging in current iteration to steps
        steps.append((lsg, mtg))
    return steps
```

Figure 4: Python code implementation for the L2H algorithm presented in Section 3. Note that we choose the aggregation function when computing the score per group as described in Appendix B.2.

## B    EXPERIMENTAL DETAILS

### B.1    DATASETS

In this work, we run experiments on four challenging vision datasets, namely CIFAR-10 and CIFAR-100 (Lake et al., 2015), Food-101 (Bossard et al., 2014) and ImageNet1K (Deng et al., 2009). CIFAR-10 and CIFAR-100 are well-established object classification datasets. The CIFAR-10 dataset consists of 60000 32x32 colored images with 32x32, divided in 10 classes: *airplane*, *automobile*, *bird*, *cat*, *deer*, *dog*, *frog*, *horse*, *ship*, *truck*. The train/test splits contain 50000 and 10000 images respectively. Similarly, also the CIFAR-100 dataset consists of 60000 32x32 colored images. However, they are organized into 100 classes. In addition, the 100 classes are grouped into 20 superclasses. As for CIFAR-10, the train/test splits also contain 50000 and 10000 images respectively. The Food101 dataset is a fine-grained classification dataset of food images, consisting of 101000 images for 101 classes. Images are high-resolution, up to 512 pixels side length. Images are split between 75750 training samples and 25250 test images. The ImageNet1K dataset, widely used in computer vision, consists of 1000 classes organized according to the WordNet hierarchy (Miller, 1995), with 1281167 training and 50000 test samples, respectively.

### B.2    IMPLEMENTATION DETAILS

For our hierarchical clustering experiments, to train the TURTLE and TEMI models on all considered datasets, we use the official code provided by the authors with recommended choices for hyperparameters (Gadetsky et al., 2024; Adaloglou et al., 2023). In particular, TEMI employs CLIPViTL/14 representations of the data, while TURTLE employs both CLIPViTL/14 and DINOv2 ViT-g/14 representations. For more details on TURTLE trained using two representation spaces, see the original paper (Gadetsky et al., 2024). We train both TEMI and TURTLE with a number of clusters $K$ equal to the true number of classes in each dataset. For each dataset, we train models on the training set, then report metrics on the test set. Note that the L2H algorithm takes as input logits from the training set to infer the hierarchy, while metrics that evaluate the quality of the hierarchy are computed on the test set. As the aggregation function $\Lambda$ in the L2H algorithm (see Section 3) we employ

$$\Lambda_{\substack{\boldsymbol{x} \in \mathcal{D}^c \\ c \in G}} g_\theta(\boldsymbol{x}) = \sum_{c \in G} \frac{1}{|\mathcal{D}^c|} \sum_{\boldsymbol{x} \in \mathcal{D}^c} g_\theta(\boldsymbol{x})$$

which we find to work well experimentally. However, other choices are possible (see also Table 4). We implement TreeVAE (Manduchi et al., 2023) with their contrastive approach using the provided PyTorch codebase with corresponding defaults. The splitting criterion is set to the number of samples, an inductive bias that benefits this baseline method, since all datasets are balanced (Manduchi et al., 2023; Vandenhirtz et al., 2024). We set the number of clusters to 10 for CIFAR-10 and to 20 for the rest, due to the computational complexity, as seen in Table 2, as well as memory complexity, since every additional leaf adds a new decoder. DeepECT (Mautz et al., 2020) is also implemented using their provided codebase with the augmented version. Note that similar to the results shown in Manduchi et al. (2023), for colored datasets, DeepECT fails to grow trees, as they always collapse, indicating that DeepECT fails to find meaningful splits. We implement agglomerative clustering using the scikit-learn library (Pedregosa et al., 2011), and fit the model using PCA embeddings of the datasets with 50 components and wards criterion (Ward, 1963) as the linkage method. Using the author's original codebase, we further train Hyperbolic Hierarchical Clustering (Liu et al., 2019) on CLIP embeddings of the respective datasets. The authors do not describe how to retrieve cluster assignments using their method, so we follow the agglomerative clustering procedure and assume the leaves of the last $k$ tree nodes created to form a cluster, where $k$ corresponds to the chosen number of clusters.

### B.3    METRICS

Here we provide more details on the metrics reported in our experiments in Section 4.1. In our comparisons, we evaluate models both on flat and hierarchical clustering.

**Flat clustering**    To assess model performance in flat clustering, for each model we take the clustering at the level of the hierarchy where the number of clusters corresponds to the true number of

classes $K$ in a given dataset. If the number of leaves at the leaf level of the hierarchy is smaller than $K$, as is the case for, e.g., TreeVAE and DeepECT on CIFAR-100, we consider the clustering at the leaf level. For flat clustering comparisons, we resort to well-established metrics, namely NMI, ARI, Accuracy, and Purity of the clusters (i.e., Leaf Purity). To compute accuracy and leaf purity, we resort to recent implementations in Gadetsky et al. (2024) and Manduchi et al. (2023), respectively.

**Hierarchical clustering** To assess the quality of a learned hierarchy, and compare the results of different models in hierarchical clustering, we resort to two metrics. Dendrogram Purity (DP), introduced in Kobren et al. (2017), extends the notion of leaf purity to evaluate the purity of hierarchical clusters, and was recently adopted to benchmark hierarchical clustering models (Manduchi et al., 2023). Following the notation of Kobren et al. (2017), let $\mathcal{C}^*$ denote the true $K$-clustering (i.e., true class labeling) of a dataset $\mathcal{D}$. Then define

$$\mathcal{P}^* = \left\{ (x_i, x_j) \forall x_i, x_j \in \mathcal{D}, x_j \neq x_j \mid C^*(x_i) = C^*(x_j) \right\}$$

as the set of pairs of data points that belong to the same true cluster. Dendrogram Purity (DP) is then defined for a hierchical clustering $\mathcal{H}$ as

$$DP(\mathcal{H}) = \frac{1}{|\mathcal{P}^*|} \sum_{k=1}^{K} \sum_{(x_i, x_j) \in \mathcal{C}_k^*} \mathrm{pur}(\mathrm{lvs}(\mathrm{LCA}(x_i, x_j)), \mathcal{C}_k^*),$$

where $\mathrm{LCA}(x_1, x_2)$ computes the least common ancestor node of data points $x_1$ and $x_2$ in $\mathcal{H}$, $\mathrm{lvs}(z)$ returns the set of leaves of the sub-tree rooted at any internal node $z$ and $\mathrm{pur}(S_1, S_2) = |S_1 \cap S_2|/|S_1|$. One possible caveat of this metric is its high correlation with Leaf Purity: with a high leaf purity, most pairs of samples sharing the true label will inevitably fall into the same leaf. To address this, we introduce an additional metric for evaluation, namely Least Hierarchical Distance. With a similar notation as above we define

$$\bar{\mathcal{P}}^* = \left\{ (x_i, x_j) \forall x_i, x_j \in \mathcal{D}, x_j \neq x_j \mid C^*(x_i) = C^*(x_j) \wedge l(x_i; \mathcal{H}) \neq l(x_j; \mathcal{H}) \right\}$$

where the function $l(x; \mathcal{H})$ returns the cluster prediction for datapoint $x$ at the leaf level of $\mathcal{H}$. Hence $\bar{\mathcal{P}}^*$ is the set of all pairs of points sharing the same true label that are *not* assigned to the same leaf in $\mathcal{H}$. Least Hierarchical Distance is then defined for a hierarchical clustering $\mathcal{H}$ as

$$LHD(\mathcal{H}) = \frac{1}{|\bar{\mathcal{P}}^*|} \sum_{(x_i, x_j) \in \bar{\mathcal{P}}^*} \frac{\log_2(td(l(x_i; \mathcal{H}), l(x_j; \mathcal{H}))) - 1}{\log_2(K) - 1}$$

where $td(l_1, l_2)$ computes the number of edges in the shortest path that connects two leaves $l_1, l_2$ in the tree defined by $\mathcal{H}$. in the tree defined by $\mathcal{H}$. Different from Dendrogram Purity, Least Hierarchical Distance only takes into consideration pairs of data points with the same true label that do not fall into the same leaf. Hence, it does not exhibit strong correlation with Leaf Purity, being more specific to the quality of the hiearchy, rather than influenced by the clustering at the leaf level.

## C ADDITIONAL RESULTS AND VISUALIZATIONS

In Table 3 we report the results for our L2H method, implemented on top of the TURTLE model, for hierarchical clustering on the ImageNet1K dataset. These results complement the ones shown in Table 1, proving that our method can reach remarkable performance for hierarchical clustering in datasets that are large in size and number of classes. Note that alternative approaches (e.g. DeepECT, TreeVAE) do not scale to a dataset of this size and number of classes.

|            | NMI ($\uparrow$) | ARI ($\uparrow$) | ACC ($\uparrow$) | LP ($\uparrow$) | DP ($\uparrow$) | LHD ($\downarrow$) |
|------------|------|------|------|------|------|------|
| L2H-TURTLE | 0.882 | 0.621 | 0.726 | 0.744 | 0.560 | 0.210 |

Table 3: Hierarchical clustering performance of our L2H method applied on top of the TURTLE pre-trained model on the ImageNet1K dataset.

In Table 4 we provide an ablation that reports the results of L2H-TURTLE on the hierarchical clustering experiments from Section 4.1 with different choices for the aggregation function $\Lambda$ in the L2H

algorithm. The results indicate that tweaks in the aggregation function alter performance, though without abrupt changes in the metrics. These results also motivate our designated choice of aggregation function—corresponding to the last row—which works well experimentally.

| | $\Lambda$ | CIFAR-10 | | CIFAR-100 | | Food-101 | |
|---|---|---|---|---|---|---|---|
| | | DP ($\uparrow$) | LHD ($\downarrow$) | DP ($\uparrow$) | LHD ($\downarrow$) | DP ($\uparrow$) | LHD ($\downarrow$) |
| | $\sum_{c \in G} \sum_{\boldsymbol{x} \in \mathcal{D}^c} g_\theta(\boldsymbol{x})$ | 0.988 | 0.258 | 0.801 | 0.244 | 0.758 | 0.294 |
| L2H-TURTLE | $\frac{1}{|G|} \sum_{c \in G} \frac{1}{|\mathcal{D}^c|} \sum_{\boldsymbol{x} \in \mathcal{D}^c} g_\theta(\boldsymbol{x})$ | 0.988 | 0.248 | 0.793 | 0.283 | 0.751 | 0.335 |
| | $\sum_{c \in G} \frac{1}{|\mathcal{D}^c|} \sum_{\boldsymbol{x} \in \mathcal{D}^c} g_\theta(\boldsymbol{x})$ | 0.988 | 0.277 | 0.803 | 0.235 | 0.758 | 0.297 |

Table 4: Results for hierarchical clustering, in terms of Dendrogram Purity and Least Hierarchical Distance, implementing the L2H algorithm with different choices for the aggregation function $\Lambda$, on top of the TURTLE model.

In Figures 5, 6, 7 we provide additional visualizations for the hierarchies obtained with our proposed method in our hierarchical clustering experiments, complementing the quantitative and qualitative evidence shown in Section 4.1. Leafs are matched to the original labels by checking the most frequent label among data points contained in the leaf. In addition to the matched label, we report purity of each leaf, in percent.

In Figure 8, we provide additional results for ImageNet with different colorings for the inferred hierarchy, supplementing our results from Section 4.2. These visualizations show where the subtree of birds (used in Figure 3b is located within the complete tree and in relation to other superclasses, such as mammals, reptiles, dogs, and clothing.

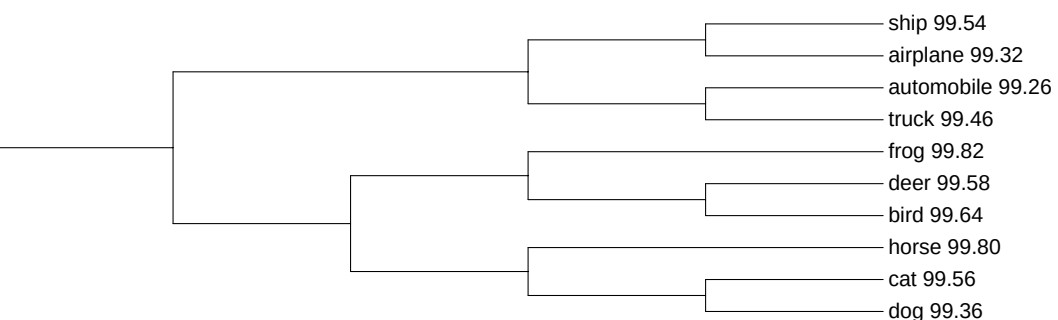

Figure 5: Visualization of the hierarchical clustering produced by L2H-TURTLE for the CIFAR-10 dataset.

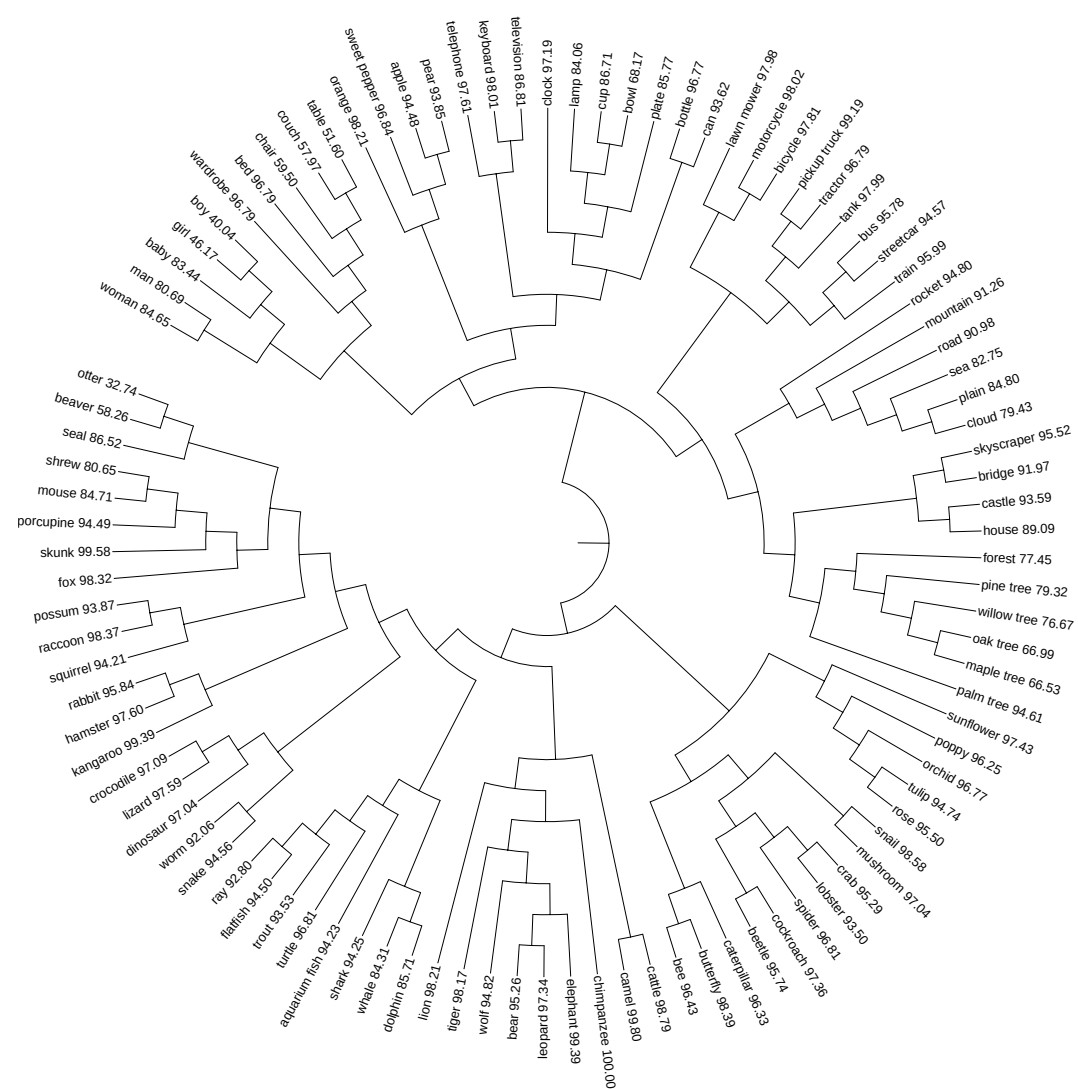

Figure 6: Visualization of the hierarchical clustering produced by L2H-TURTLE for the CIFAR-100 dataset.

918
919
920
921
922
923
924
925
926
927
928
929
930
931
932
933
934
935
936
937
938
939
940
941
942
943
944
945
946
947
948
949
950
951

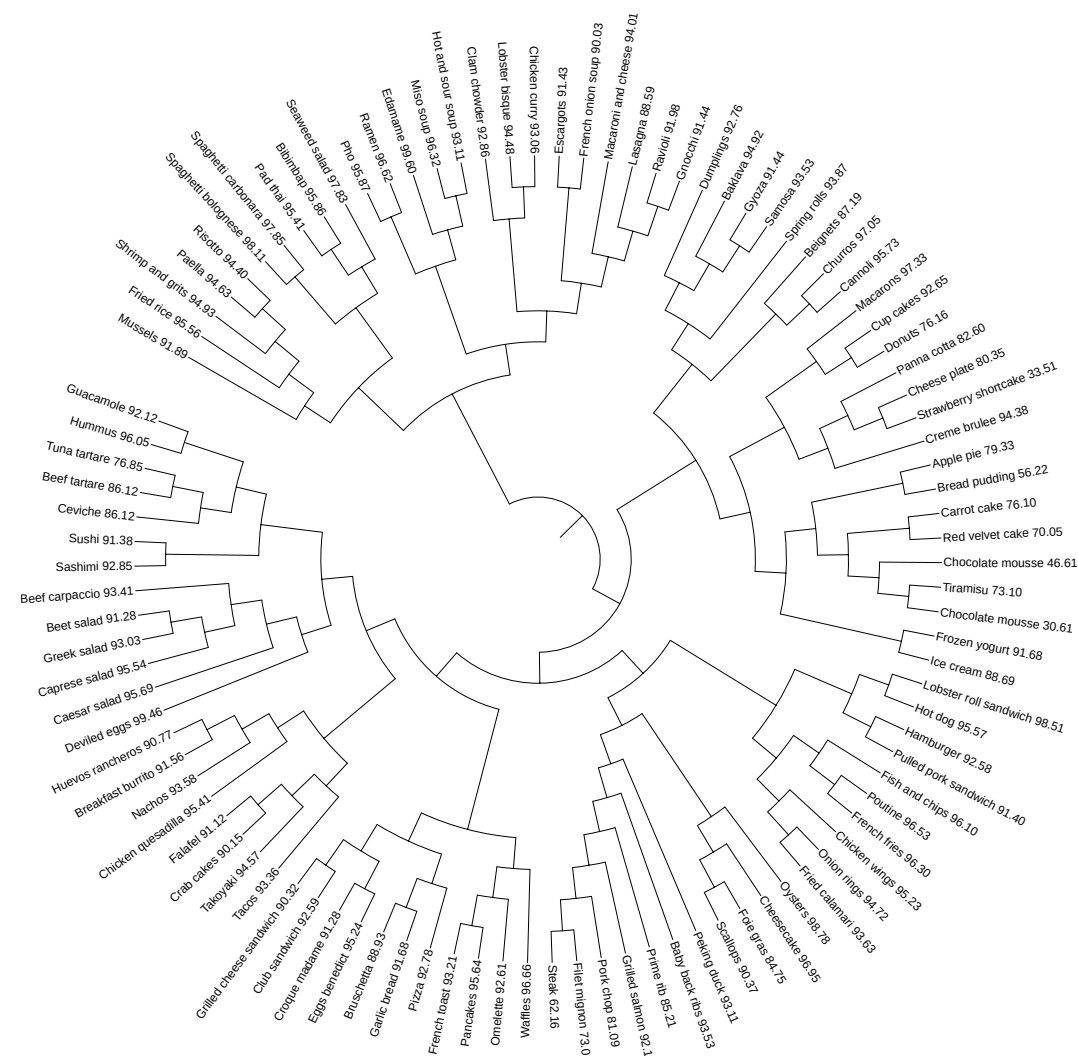

Figure 7: Visualization of the hierarchical clustering produced by L2H-TURTLE for the Food-101 dataset.

952
953
954
955
956
957
958
959
960
961
962
963
964
965
966
967
968
969
970
971

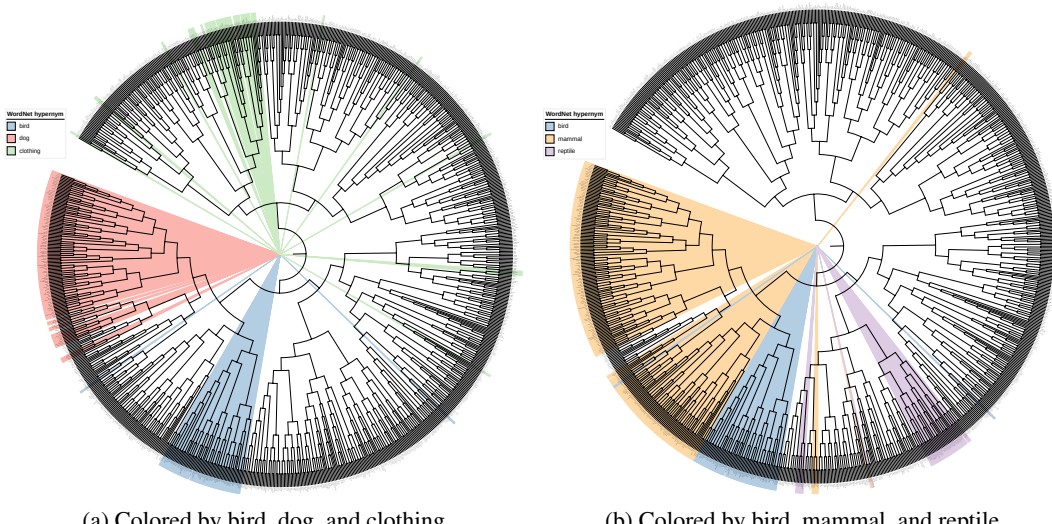

(a) Colored by bird, dog, and clothing    (b) Colored by bird, mammal, and reptile

Figure 8: Visualization of the hierarchical clustering produced by L2H-TURTLE for the ImageNet-1K dataset. We show the complete tree of 1K classes colored by the corresponding WordNet hypernyms "bird", "dog", and "clothing" (Figure 8a) and by "bird", "mammal", and "reptile" (Figure 8b).

