# OpenReview forum: "From Logits to Hierarchies: Hierarchical Clustering made Simple"
_ICLR.cc/2025/Conference — Submitted to ICLR 2025_

### Official Review · Reviewer_abpg · 2024-11-02

**Soundness:** 2
**Presentation:** 1
**Contribution:** 2
**Rating:** 5
**Confidence:** 3

**Summary:**

This paper proposes a simple method for hierarchical clustering based on pre-trained non-hierarchical models. Specifically, the method performs on top of (non-hierarchical) pre-trained models, which uses the logits output by the pre-trained models to iteratively compute clusters in a fine-to-coarse manner. The authors verify the effectiveness of the method on four datasets, including CIFAR-10, CIFAR-100, Food-101 and ImageNet1K.

**Strengths:**

The proposed method is simple.
The proposed method looks effective from the experimental results.

**Weaknesses:**

+ The paper presents some challenges in clarity, particularly in the methods section, which may hinder reviewers from fully understanding how the proposed method operates. For example,
  - The terms "clusters" and "groups" seem to be used somewhat interchangeably, which can lead to confusion regarding their meanings. It would be helpful for the authors to clarify the distinctions between these terms in the context of the paper.
  - it should number the equations and reference specific formulas in the algorithm table to illustrate the execution process of the proposed method more effectively.

+ The authors assert that the proposed method does not require fine-tuning of the pre-trained model. However, it would be helpful to clarify how the logits are generated for different datasets. For example, in a network trained on ImageNet1K, the output is 1000-dimensional. If we are now clustering on CIFAR-10, does this imply that its logits are also 1000-dimensional? In other words, does this mean that the number of clusters at the finest-level hierarchy is 1,000?

+ In the context of hierarchical clustering, determining the number of hierarchies is crucial. I would appreciate further explanation on how this is accomplished within the proposed method.

+ Regarding line 190, where $s(G)$ represents the sum of the probabilities of samples belonging to $G$, it seems that if $G$ has more samples, its $s(G)$ value would naturally be larger. This raises the question of whether the relatedness between clusters is sensitive to the number of samples, which may not align with our intuitive understanding.

**Questions:**

Please refer to the weakness section.

---

> ### Author Response · Authors · 2024-11-22
> **Reply to Reviewer abpg**
>
> We thank the Reviewer for praising the simplicity and effectiveness of our method. However, we are surprised by such a low score on presentation, and hope that we can best address the concerns by the Reviewer in this regard, in particular as clarity of the paper was praised by other Reviewers. We address below each question and concern raised by the Reviewer, incorporating useful suggestions.
>
> > For example, - The terms "clusters" and "groups" seem to be used somewhat interchangeably, which can lead to confusion regarding their meanings. - It would be helpful for the authors to clarify the distinctions between these terms in the context of the paper.
>
> We thank the Reviewer for the suggestion, and we clarify this better in the updated manuscript. As specified in its description in the paper, our algorithm iteratively groups the clusters from a pre-trained model to build a hierarchy. Hence *group* refers to a set of clusters that are grouped together at a given step of the procedure, while *cluster* is used to refer to a given cluster found by the pre-trained model. To highlight this distinction, we use two different terms.
>
> > It should number the equations and reference specific formulas in the algorithm table to illustrate the execution process of the proposed method more effectively.
>
> We thank the Reviewer for the suggestion, which we have incorporated in the updated manuscript.
>
> > The authors assert that the proposed method does not require fine-tuning of the pre-trained model. However, it would be helpful to clarify how the logits are generated for different datasets. For example, in a network trained on ImageNet1K, the output is 1000-dimensional. If we are now clustering on CIFAR-10, does this imply that its logits are also 1000-dimensional? In other words, does this mean that the number of clusters at the finest-level hierarchy is 1,000?
>
> As a clarification, the pre-training of the flat model (e.g. TURTLE) is done on the given dataset we consider in each experiment, and for each dataset we specify a corresponding number of clusters $K$. Note that for the results in section 4.1 TURTLE and TEMI are fully unsupervised pre-trained models, while for results in Section 4.2 InternImage is pre-trained with supervision. Finally, again to clarify, the number of leafs in the hierarchy obtained with our method corresponds to the number of clusters $K$ set for the pre-trained model. However, an advantage of hierarchical clustering compared to flat clustering is that by inspecting the hierarchy one can obtain multiple clusterings at different levels of granularity.
>
> > In the context of hierarchical clustering, determining the number of hierarchies is crucial. I would appreciate further explanation on how this is accomplished within the proposed method.
>
> Here we are unsure of what the Reviewer means by "number of hierarchies". Could the Reviewer please clarify this?
>
> > Regarding line 190, where $s(G)$ represents the sum of the probabilities of samples belonging to $G$, it seems that if $G$ has more samples, its s(G) value would naturally be larger. This raises the question of whether the relatedness between clusters is sensitive to the number of samples, which may not align with our intuitive understanding.
>
> Note that our choice of aggregation function (Appendix B.2, line 732) includes a normalization factor by the size of each cluster in the group, hence accounting for relative size of clusters in terms of number of samples. In addition, we have now included an ablation in Appendix C of the updated manuscript that shows the impact of changes in the aggregation function on the hierarchical clustering metrics.
>
> We have addressed the Reviewer's concerns and questions, providing useful clarifications, and we hope this will result in an increase in the score. We remain available to address any further questions or concerns.

---

> > ### Comment · Reviewer_abpg · 2024-11-23
> >
> > Thank you for your response; my concerns have been partially addressed.
> >
> > + The following question remains unclear: If a network pre-trained on ImageNet1K is used for clustering on a new dataset, does this imply that its logits are 1000-dimensional? In other words, does this mean that the number of clusters at the finest level of the hierarchy corresponds to 1,000?
> >
> > + Regarding the term "number of hierarchies," it refers to the number of levels or tiers in the clustering hierarchy. Alternatively, as shown in your Figure 2, the depth of the tree can be interpreted as the number of hierarchies.

---

> ### Author Response · Authors · 2024-11-24
> **Reply to Reviewer abpg**
>
> We thank the Reviewer for the reply, and are happy to have addressed part of the concerns.
> We address below the remaining questions:
> > The following question remains unclear: If a network pre-trained on ImageNet1K is used for clustering on a new dataset, does this imply that its logits are 1000-dimensional? In other words, does this mean that the number of clusters at the finest level of the hierarchy corresponds to 1,000?
>
> Yes, the understanding from the Reviewer is correct. If one uses a flat clustering model pre-trained on ImageNet1K for clustering on a new dataset -- provided that $K$ was set to the true number of classes of ImageNet1K for pre-training -- the logits will be 1000-dimensional. Hence implementing our algorithm on top would result in a hierarchy with 1000 clusters at the lowest level (i.e. 1000 leafs). However, the hierarchy can be inspected for clusterings at multiple granularities. For instance if a 1000-clustering is thought to be an over-clustering for the new dataset, highest levels of the hierarchy can yield more useful clusterings.
>
> > Regarding the term "number of hierarchies," it refers to the number of levels or tiers in the clustering hierarchy. Alternatively, as shown in your Figure 2, the depth of the tree can be interpreted as the number of hierarchies.
>
> The depth of the hierarchy obtained with our method is influenced by two factors. The first factor is the number of clusters $K$ chosen for the pre-trained model. The larger $K$, the larger the number of leaves in the hierarchy, and hence the depth to which the tree can grow. The second factor, given that the depth of the tree is not univocally determined by the number of leaves, is the nature of the hierarchical relations between clusters that characterize the data and are modelled by our approach. Note that choosing the right number of clusters $K$ is relevant when dealing with flat clustering approaches, and most methods assume that the true number of clusters (or a proxy for it) is known. With hierarchical clustering approaches like ours, $K$ is also a relevant hyperparameter, but the requirement of specifying a correct value a-priori for $K$ is alleviated with respect to the flat case, as inspecting the hierarchy can yield clusterings at different levels of granularity [3].
>
>  [3] Chami et al. From trees to continuous embeddings and back: Hyperbolic hierarchical clustering. In Advances in Neural Information Processing Systems, 2020.
>
> We have addressed the Reviewer's remaining questions, and we hope this will result in an increase in the score. We remain available to address any further questions or concerns.

---

> > ### Comment · Reviewer_abpg · 2024-11-26
> >
> > I appreciate the author's further response. I have increased my score to 5; however, I still believe that the paper is not yet ready for publication at ICLR. The author should further improve the clarity of the writing and provide a more thorough validation of the method's effectiveness.

---

### Official Review · Reviewer_k4QD · 2024-11-04

**Soundness:** 2
**Presentation:** 3
**Contribution:** 2
**Rating:** 3
**Confidence:** 4

**Summary:**

The authors propose a new hierarchical clustering algorithm that leverages a pre-trained flat clustering models.
The key observation is that the logits of those model encode class similarity.

**Strengths:**

- The paper is well-written and easy to read.
- The authors compare their method with three related methods.
- The authors provide the pseudo code of their algorithm.

**Weaknesses:**

The contribution is rather limited and incremental. It is well known that the logits encode class similarity, and have been frequently used to reconstruct classification hierarchy. Examples for previous work with similar observation:
- https://arxiv.org/abs/2007.06068
- https://dl.acm.org/doi/abs/10.1145/3491102.3501823

The evaluation is based on two pre-trained image clustering models, TURTLE and TEMI.
This limits the generalizability of the results, as these are not the same backbones used for the other baselines.

Minor presentation remarks:
- Provide full names of the acronyms NMI and ARI
- datapoints => data points

**Questions:**

Did you consider using the logits of standard image classification models (e.g. a ResNet trained to classify ImageNet)?

---

> ### Author Response · Authors · 2024-11-22
> **Reply to Reviewer k4QD**
>
> We thank the Reviewer for praising the writing of our paper, as well as our proposed baselines.
> We address below each question and concern raised by the Reviewer, incorporating useful suggestions.
>
> > The contribution is rather limited and incremental. It is well known that the logits encode class similarity, and have been frequently used to reconstruct classification hierarchy.
>
> We agree with the Reviewer that the idea of exploiting information in the logits to compute class similarity has been explored in classification. However, to the best of our knowledge, we are the first ones to use the same intuition for hierarchical clustering. Additionally, we couple this intuition with the idea of moving away from costly ad-hoc hierarchical approaches, and focusing on a lightweight adaptation of pre-trained flat models to hierarchical clustering tasks. Our work puts existing deep hierarchical clustering models in perspective, showing that they are markedly surpassed by our proposed alternative strategy in terms of both performance and efficiency. Overall, we believe our work provides valuable insights for both researchers and practitioners interested in hierarchical clustering tasks.
>
> > The evaluation is based on two pre-trained image clustering models, TURTLE and TEMI. This limits the generalizability of the results, as these are not the same backbones used for the other baselines.
>
> Again we'd like to remark that a key contribution of this work is introducing a different perspective on hierarchical clustering compared to previous works, and proposing to efficiently adapt a pre-trained flat model to obtain a hierarchy of clusters, rather than designing costly and complex ad-hoc hierarchical approaches.
>
> > Minor presentation remarks:
> > Provide full names of the acronyms NMI and ARI
> datapoints => data points
>
> Thank you for these useful suggestions/remarks, we updated the manuscript accordingly.
>
> > Questions:
> > Did you consider using the logits of standard image classification models (e.g. a ResNet trained to classify ImageNet)?
>
> We are not sure here what the Reviewer precisely means with "standard" image classification models. In section 4.2, we provide results for our method applied using the logits of a pre-trained ImageNet1K classifier, namely InternImage [4].
>
> We have addressed the Reviewer's concerns and questions, clarifying the value of our contributions and results in this work, and we hope this will result in an increase in the score. We remain available to address any further questions or concerns.

---

### Official Review · Reviewer_LaeG · 2024-11-04

**Soundness:** 3
**Presentation:** 3
**Contribution:** 2
**Rating:** 3
**Confidence:** 3

**Summary:**

The authors provide a simple and fast hierarchical clustering method, which works well in the case of many-class problems. It works on the pretrained representation in the logits space. The method is a version of aglomerative clustering which can be based on a chosen standard clustering backbone. The paper is well written. The idea is simple and can be easily applied even to large datasets.

**Strengths:**

The paper is clearly written, the experiments are properly conducted and shows that the method outperforms other hierarchical approaches. The idea is nice as it uses as a backbone a non-hierarchial clustering method.

**Weaknesses:**

The novelty of the approach is rather limited, the method is in fact a shallow clustering approach applied in the representation space. Consequently, since in training the representation the knowledge of classes was used, it is hard to say how the method would deal in the case when the model did not know the proper classes beforehand. In particular a test of themodel on a few common unsupervised representations on ImageNet is necessary (for example given by SimClr or MaskedAutoEncoders).

**Questions:**

It is necessary to validate the approach on representation space constructed by unsupervised models.

---

> ### Author Response · Authors · 2024-11-22
> **Reply to Reviewer LaeG**
>
> We thank the Reviewer for praising the idea behind our work, the soundness of our experiments, and the clarity of our paper.
>
> We address below each question and concern raised by the Reviewer.
>
> > The novelty of the approach is rather limited, the method is in fact a shallow clustering approach applied in the representation space. Consequently, since in training the representation the knowledge of classes was used, it is hard to say how the method would deal in the case when the model did not know the proper classes beforehand. In particular a test of themodel on a few common unsupervised representations on ImageNet is necessary (for example given by SimClr or MaskedAutoEncoders).
>
> There seems to be some confusion here. In Section 4.1 the results for our method (L2H-TEMI, L2H-TURTLE) are obtained *without any knowledge of the true labels during training*. Hence in a fully unsupervised setup. Only in Section 4.2 the pre-trained model (i.e. InternImage [4]) is a supervised one, as this section shows that our approach is general, and can also be applied to supervised setups. Finally note that we have now provided the results for our method on ImageNet1K in Appendix C of the updated manuscript, again in a fully unsupervised setup.
>
> > It is necessary to validate the approach on representation space constructed by unsupervised models.
>
> As clarified above, for all results in section 4.1, the setup is fully unsupervised, both when training the backbone model (TURTLE or TEMI) and when applying our method on top.
>
> We have addressed the Reviewer's concerns and questions, and clarified that our approach was tested on challenging fully unsupervised setups achieving remarkable performance. We hope that this will result in an increase in the score. We remain available to address any further questions or concerns.

---

### Official Review · Reviewer_AeLC · 2024-11-04

**Soundness:** 3
**Presentation:** 3
**Contribution:** 2
**Rating:** 5
**Confidence:** 4

**Summary:**

This work proposes a new algorithm for generating hierarchies based on logits of pre-trained models. The new method outperforms other deep clustering techniques both in terms of performance and runtime.

**Strengths:**

The methods elegantly bypass the standard O(N^2) complexity required for standard agglomerative clustering

The model outperforms the other baselines both in terms of runtime and clustering performance.

The paper is well written and easy to follow.

**Weaknesses:**

I'm concerned that the suggested models received a better starting point. If I understand correctly, the pre-trained models know better how to cluster to the starting 10/100/101 clusters. It would benefit this work if there would be an ablation study that demonstrated the good results caused by the suggested algorithm and not by the pre-trained models.

The suggested algorithm is generic, how does it perform on non-vision-based clustering tasks?
How does it perform compared to other (more) deep hierarchical models?
I hope to see a broader comparison, with more than two models for deep custring and on various clustering tasks aside from vision.

The runtime comparison is not 100% fair, in my opinion, since L2H utilizes pre-trained models.
The pre-trained models already know how to cluster the leafs, while other models need to learn it.

**Questions:**

Do other methods assume a starting division to K clusters?

---

> ### Author Response · Authors · 2024-11-22
> **Reply to Reviewer AeLC**
>
> We thank the Reviewer for praising the elegance and performance of our approach, as well as the writing of our paper.
>
> We address below each question and concern raised by the Reviewer, taking the opportunity to provide useful clarifications.
>
> > I'm concerned that the suggested models received a better starting point. If I understand correctly, the pre-trained models know better how to cluster to the starting 10/100/101 clusters. It would benefit this work if there would be an ablation study that demonstrated the good results caused by the suggested algorithm and not by the pre-trained models.
>
> Leveraging the knowledge from pre-trained flat clustering models to efficiently perform hierarchical clustering is an intrinsic feature of our approach. In fact, we propose a novel perspective on hierarchical clustering compared to recent works. We argue and demonstrate that, instead of designing complex ad-hoc approaches, a simple plug-in method can extend a flat model to produce a hierarchy, reaching much better performance and much higher efficiency.
>
> > The suggested algorithm is generic, how does it perform on non-vision-based clustering tasks? How does it perform compared to other (more) deep hierarchical models? I hope to see a broader comparison, with more than two models for deep custring and on various clustering tasks aside from vision.
>
> As argued in recent work [1],  only a few deep learning based methods for hierarchical clustering have been proposed in recent years. We followed recent relevant work to select baselines [1], and even added HypHC [3]. Note that as highlighted in [1] some related works do not provide publicly available code. Testing our approach on non-vision realms is definitely interesting, and we will keep this suggestion for future work.
>
> > The runtime comparison is not 100% fair, in my opinion, since L2H utilizes pre-trained models. The pre-trained models already know how to cluster the leafs, while other models need to learn it.
>
> Note that the L2H-TURTLE entry in Table 2 reports the time needed to train the TURTLE model plus the time needed to run our L2H algorithm on top. Hence, comparing this value with the run time of the other models constitutes a fair comparison, and demonstrates the much higher efficiency of our approach.
>
> > Questions:
> > Do other methods assume a starting division to K clusters?
>
> No, leveraging a pre-trained flat $K$-clustering model to efficiently produce a hierarchy, rather than designing costly ad-hoc hierarchical approaches, is the idea we explore in this work. Note as well that deep hierarchical clustering models such as TreeVAE[1] and DeepECT[2] are divisive, i.e. they build a tree from the root in a top-down fashion.
>
> We have addressed the concerns and questions from the Reviewer, and hope that this will result in an increase in the score. We also remain available to address any additional questions or concerns.

---

> > ### Comment · Reviewer_AeLC · 2024-11-24
> > **Official Comment by Reviewer AeLC**
> >
> > Thank you for your response. Since an ablation study and a broader comparison were not performed, I would like to keep my score.

---

### Official Review · Reviewer_z6t7 · 2024-11-06

**Soundness:** 2
**Presentation:** 3
**Contribution:** 2
**Rating:** 3
**Confidence:** 4

**Summary:**

The paper presents a method to map from the logit space output from a pretrained clustering network to convert its output into a hierarchical structure. They demonstrate that by building on existing foundation models and non-hierarchical clustering methods, this simple method can outperform bespoke deep hierarchical clustering methods.

**Strengths:**

It is valuable to have strong baselines for complex methods, even when those methods are simple.

The paper is generally well written.

**Weaknesses:**

1. Experiments are too restricted. In the introduction, the authors state that much real-world data is hierarchical, such as taxonomic data. However, they barely run experiments on datasets that possess hierarchical labels. CIFAR-10 does not have hierarchical labels, and so it is unclear how the authors evaluated it for dendritic purity, etc. As far as I am aware, Food101 doesn't either. CIFAR-100 only has one intermediate level in its hierarchy. They refer to the datasets used as being challenging, but I don't think it is appropriate to describe CIFAR-10 as such when we have models achieving >99% accuracy on it. They refer to CIFAR-100 and Food101 as having a large number of classes, but given that models are routinely trained on IN-1k (1,000 classes), IN-21k (21,000 classes), iNaturalist (10,000 classes), and 100 classes is quite small in comparison to these datasets. For work that is benchmarking a relatively simple method, I think it is important that the authors benchmark on more complex and more real-world datasets such as iNaturalist or BIOSCAN-5M from the taxonomic labelling perspective, or even non-image modalities for [more comprehensive analysis]. For a method that trains in 5 minutes on IN-1k, I do not think the authors can justify not exploring larger datasets and demonstrating the methodology works well at scale, both in terms of its performance and execution time. Even worse though, quantitative IN-1k results are not shown (i.e. in Table 1).

2. Baselines are inadequate. What is being proposed in the paper is a method for mapping from logits to a hierarchy, harnessing a couple of methods (TEMI and TURTLE) which can map from the embeddings of pretrained encoder (CLIP and DINOv2) to logits for clusters.
(2a.) The authors introduce the flexibility in their algorithm to consider multiple aggregation operations, but only consider summation ($Λ=\Sigma$). This leaves the question of what is the best aggregation operation unclear.
(2b.) Moreover, the paper leaves open the question of whether this is a *good* methodology of mapping from the TEMI or TURTLE outputs to a hierarchy. They present one method for this mapping, but there are no comparators given for it. For example, what if you used agglomerative clustering to cluster the TEMI or TURTLE logits? Does the L2H method beat this obvious baseline?
(2c.) The L2H-TEMI/TURTLE methods presented have the benefit of building on pretrained foundation models, whereas the competing methods are trained from scratch. Hence it would perhaps be more appropriate to compare against alternative methods for producing clusters from the embeddings of pretrained models, for instance [Lowe et. al. (2024)](https://arxiv.org/abs/2406.02465) investigate methodologies for doing this. [That paper concludes that the best way to cluster embeddings from a pretrained model is UMAP with >5 dims for dimensionality reduction, followed by Agglomerative clustering. So although they don't analyze hierarchical embeddings, the methodology they recommend is actually hierarchical and should serve as a useful baseline for this paper.]

3. Algorithm 1 (which is in a sense the main output of the work) is not adequately well written.
    - The same variable, $s$, is used for both the step number and the score.
    - Some variables are defined in the text of the paper, and used in the algorithm without definitions in the algorithm itself. The algorithm should stand alone without needing to read the rest of the paper to understand it. $f_θ$ is defined and never used, whilst $g_θ$, $g^m_θ$, $K$, $Λ$ are used and never defined.
    - argmin yields a single index, not a set, so it is unclear why the authors use an $\in$ symbol in $G^*\in \text{argmin}_{G\in \mathrm{G}} s(G)$, etc. Perhaps this is to cover the edge case of ties..? But this unnecessarily adds confusion to the notation.
    - $K$ is not updated in the outer loop, so the inner loop at L173 refers to cluster indices that no longer correspond to unmerged clusters.

**Typos and minor points**
- L311 Agglomerative clustering is abbreviated as Agg, but as far as I can see, this abbreviation is never used.
- The methodology for several things is only given in the appendix (e.g. agglomerative clustering), but nowhere in the main text does it let the reader know this or where in the appendix one might find these details (which a reader otherwise may well assume are omitted from the text entirely).
- There are a couple of places where a word is repeated (e.g. L271 "Appendix Appendix")
- L730 The text says it is giving the aggregation function, $Λ$, but the equation is the score function
- L734 "possiblle"
- L341 British English "modelled" contrasts with the American English used in the rest of the text
- Some references are cased incorrectly, e.g. key reference "Deepect" -> DeepECT
- Some references don't include publication details, e.g. Bengio (2014)
- Some references cite arXiv versions of papers where peer-reviewed versions are available (which should generally be preferred for citations) e.g. Karthik (2021b)
- Most references don't have links to the paper being cited, which should ideally be present for the convenience of the reader in the modern, digital era. DOIs can be made clickable simply by importing the doi package in main.tex (and don't need a URL field also present in the bibtex)
- One reference has mojibake: Nguyen (2024) "Identification of distinct subgroups of sj&#xf6;gren’s disease"

**References**
- iNaturalist-21 [arXiv:2103.16483](https://arxiv.org/abs/2103.16483)
- BIOSCAN-5M: [arXiv:2406.12723](https://arxiv.org/abs/2406.12723)
- Lowe et. al. (2024): [arXiv:2406.02465](https://arxiv.org/abs/2406.02465)

[N.B I realized a sentence was half-formed after the reviews were released, and added it later in an update to the review.]

**Questions:**

Q1. Why is IN-1k not included in Table 1?

Q2. What is the input to the agglomerative clustering baseline? Is it the raw pixel values of the images?

Q3. Is the performance of L2H-TEMI on flat clustering metrics identical to the performance of TEMI? Similarly for (L2H-)TURTLE.

Q4. Did you really train the auto-encoder for DeepECT from scratch on CIFAR-10/100 in 25 minutes on a single CPU core...? This seems rather unlikely, but it does appear to be what the paper claims.

Q5. In Fig 2, why is there a space between woman and otter but not between palm tree and sunflower? The two pairs are equally far from each other.

---

> ### Author Response · Authors · 2024-11-22
> **Reply to Reviewer z6t7 (part 1)**
>
> We thank the Reviewer for praising the writing of our paper, and the value of our work as a strong baseline to benchmark future methods for hierarchical clustering. We address below each question and concern raised by the Reviewer, and consider and incorporate valuable suggestions.
>
> > 1. Experiments are too restricted. In the introduction, the authors state that much real-world data is hierarchical, such as taxonomic data [...]
>
> Our choice of datasets for our hierarchical clustering experiments in this paper (CIFAR-10, CIFAR-100, Food-101) is backed up by a number of relevant considerations. Existing deep hierarchical clustering methods (e.g. TreeVAE[1], DeepECT[2]) already heavily struggle on these datasets, as clearly shown in our results. Note that the difficulty level of a dataset is considered in the context of a given task, and we prove in our paper that the chosen datasets are indeed markedly challenging for existing hierarchical clustering approaches. Showing that our method can achieve remarkable results on these datasets already represents a substantial improvement over existing methods. Note that, if we had chosen larger datasets for our experiments in Section 4.1, it would have not been possible to compare our model with alternative methods (that cannot scale to such settings), which was the main goal. In addition, it is clear that if current approaches cannot tackle the proposed settings, there is no hope of them working on more challenging datasets. This strongly speaks to the relevance of proposing an alternative strategy to ad-hoc complex deep hierarchical clustering models---since they prove to fail when applied to complex setups---which is what we do in this work. We did not include ImageNet1K as a dataset in Section 4.1 as all models except for ours cannot tackle this dataset due to scalability issues. However, we now provide the results for L2H-TURTLE on ImageNet1K in the updated manuscript (Appendix C), that demonstrate that our method performs well also on large complex datasets with a high number of classes. On a final note, we'd like to clarify that dendrogram purity does not need hierarchical labels to be computed, and hence can be evaluated on datasets with only flat labels available.
>
> > 2. Baselines are inadequate. What is being proposed in the paper is a method for mapping from logits to a hierarchy, harnessing a couple of methods (TEMI and TURTLE) which can map from the embeddings of pretrained encoder (CLIP and DINOv2) to logits for clusters. (2a.) The authors introduce the flexibility in their algorithm to consider multiple aggregation operations, but only consider summation (). This leaves the question of what is the best aggregation operation unclear. (2b.) Moreover, the paper leaves open the question of whether this is a good methodology of mapping from the TEMI or TURTLE outputs to a hierarchy. They present one method for this mapping, but there are no comparators given for it. For example, what if you used agglomerative clustering to cluster the TEMI or TURTLE logits? Does the L2H method beat this obvious baseline? (2c.) The L2H-TEMI/TURTLE methods presented have the benefit of building on pretrained foundation models, whereas the competing methods are trained from scratch. Hence it would perhaps be more appropriate to compare against alternative methods for producing clusters from the embeddings of pretrained models, for instance Lowe et. al. (2024) investigate methodologies for doing this. [That paper concludes that the best way to cluster embeddings from a pretrained model is UMAP with >5 dims for dimensionality reduction, followed by Agglomerative clustering. So although they don't analyze hierarchical embeddings, the methodology they recommend is actually hierarchical and should serve as a useful baseline for this paper.]
>
> In this paper, we challenge the perspective of recent work on deep hierarchical clustering that focuses on complex ad-hoc and computationally expensive approaches. To do so, we provide a concrete example of a novel, simple and efficient method to perform hierarchical clustering based on a pre-trained flat clustering model. We compare with state-of-the-art deep hierarchical clustering models--basing our selection of baselines also on relevant recent work [1]--and therefore we argue that we provide adeuquate baselines to validate the effectiveness of our approach.
>
> 2a.) Here there seems to be some confusion. As aggregation function we use the function presented in Appendix B.2 (line 732), which we find to work well experimentally. However, we now included an ablation in Appendix C of the updated manuscript, showing the effect of changes in the aggregation function on the hierarchical clustering performance.
>
>
> **[Note: this reply continues in a separate comment.]**

---

> ### Author Response · Authors · 2024-11-22
> **Reply to Reviewer z6t7 (part 2)**
>
> 2b.) We thank the Reviewer for the suggestion. However, the suggested baseline is derived from the idea we present in this work, i.e. to leverage the logits of pre-trained flat clustering models for efficient hierarchical clustering, as an alternative to costly ad-hoc approaches. We remark that to the best of our knowledge we are the first ones to explore this strategy for efficiently solving hierarchical clustering tasks.
>
> 2c.) We thank the Reviewer for the valuable reference, that we will include in the camera-ready version. We were not aware of this concurrent work. However, as stated above, our main goal here is to propose a lightweight solution to challenge the necessity for costly ad-hoc deep hierarchical clustering methods. Hence we do not opt for including it as a baseline, also taking into account the fact that it is concurrent work.
>
>
> > Algorithm 1 (which is in a sense the main output of the work) is not adequately well written.
>
> > The same variable, $s$, is used for both the step number and the score.
>
> We see how this can be confusing, and we adapted the algorithm to avoid this.
>
> > Some variables are defined in the text of the paper, and used in the algorithm without definitions in the algorithm itself. The algorithm should stand alone without needing to read the rest of the paper to understand it. $t_\theta$ is defined and never used, whilst $g_\theta,g^m_\theta, K, \Lambda$ are used and never defined.
>
> We believe that presenting an algorithm in the context of a paper should prioritize clarity and readability, rather than optimizing for it being self-contained. We believe that step-by-step definitions of functions $f_{\theta},g_{\theta},h_{\theta},f^m_{\theta},g^m_{\theta},h^m_{\theta}$ in the main text help the reader understand our idea and method. Referring to these function definitions in the algorithm allows us to provide a concise and readable outline of our procedure that is easy to grasp. However, we thank the Reviewer for the suggestion, and we have added details to our algorithm to be more specific in how our notation is defined.
>
> > argmin yields a single index, not a set, so it is unclear why the authors use an  symbol in , etc. Perhaps this is to cover the edge case of ties..? But this unnecessarily adds confusion to the notation.
>
> Exactly, the $\in$ symbol is for mathematical correctness, since there might be ties. However we consider the suggestion from the Reviewer, and replace it with $=$ to not overload notation.
>
>
> > $K$ is not updated in the outer loop, so the inner loop at L173 refers to cluster indices that no longer correspond to unmerged clusters.
>
> Here there seems to be some confusion. Note that $K$ should not be updated, and at each iteration all clusters (except for the ones in $G^*$) should be considered. If not convinced by the correctness of the algorithm one can confront it with the Python snippet we provide, which can be executed.
>
>
> > Typos and minor points
>
> > L311 Agglomerative clustering is abbreviated as Agg, but as far as I can see, this abbreviation is never used
>
> Thanks for spotting this, we removed "(Agg)".
>
> > The methodology for several things is only given in the appendix (e.g. agglomerative clustering), but nowhere in the main text does it let the reader know this or where in the appendix one might find these details (which a reader otherwise may well assume are omitted from the text entirely).
>
> We make several references to the Appendix (e.g. lines 215, 262, 271), but we take the suggestion from the Reviewer, and make it more explicit in the updated manuscript that the full details on experiments can be found in the Appendix.
>
> > There are a couple of places where a word is repeated (e.g. L271 "Appendix Appendix")
> > L730 The text says it is giving the aggregation function $\Lambda$, but the equation is the score function
> > L734 "possiblle"
> > L341 British English "modelled" contrasts with the American English used in the rest of the text
> > Some references are cased incorrectly, e.g. key reference "Deepect" -> DeepECT
> > Some references don't include publication details, e.g. Bengio (2014)
> > Some references cite arXiv versions of papers where peer-reviewed versions are available (which should generally be preferred for citations) e.g. Karthik (2021b)
> > One reference has mojibake: Nguyen (2024) "Identification of distinct subgroups of sjögren’s disease"
>
> Thank you for pointing these typos out, we have corrected them.
>
>
> > Most references don't have links to the paper being cited, which should ideally be present for the convenience of the reader in the modern, digital era. DOIs can be made clickable simply by importing the doi package in main.tex (and don't need a URL field also present in the bibtex)
>
> To the best of our knowledge, at least in top-tier ML conferences, it is rather customary not to do so. We therefore thank the Reviewer for the suggestion, but do not opt for doing so.
>
> **[Note: this reply continues in a separate comment.]**

---

> ### Author Response · Authors · 2024-11-22
> **Reply to Reviewer z6t7 (part 3)**
>
> > Questions:
> Q1. Why is IN-1k not included in Table 1?
>
> As explained above, we do not include ImageNet1K in Table 1 one because deep hierarchical baselines (e.g. DeepECT[2], TreeVAE[1]) cannot scale to the size and number of classes of this dataset. For example, TreeVAE trains a specific decoder for each leaf and hence training it with a large number of leaves (i.e. clusters) is infeasible.
>
> > Q2. What is the input to the agglomerative clustering baseline? Is it the raw pixel values of the images?
>
> Yes, we train the agglomerative clustering baseline on the data space.
>
> > Q3. Is the performance of L2H-TEMI on flat clustering metrics identical to the performance of TEMI? Similarly for (L2H-)TURTLE.
>
> Yes, and it is so by construction of our method.
>
> > Q4. Did you really train the auto-encoder for DeepECT from scratch on CIFAR-10/100 in 25 minutes on a single CPU core...? This seems rather unlikely, but it does appear to be what the paper claims.
>
> We want to clarify that we do not make this claim in the paper. DeepECT[2] is trained on a GPU for the results in Table 2. We state instead that, with logits from a pre-trained flat model as input, our algorithm performs hierarchical clustering on ImageNet-sized datasets in a few minutes on a single CPU core.
>
>
> > Q5. In Fig 2, why is there a space between woman and otter but not between palm tree and sunflower? The two pairs are equally far from each other.
>
> This is merely an effect of the visualization tool we are using.
>
> We have addressed each question and concern from the Reviewer, and clarified the scope, extensiveness, and significance of our contributions and results, and we hope that this will results in an increase in the score. In case of any residual questions or concerns, we remain open to address them.

---

> ### Comment · Reviewer_z6t7 · 2024-11-26
>
> I thank the authors for addressing some of my concerns, however most are not resolved to my satisfaction so I will keep my score.
>
> I remain concerned that the method about the lack of baselines to compare against.
> (2b) The authors propose a methodology to map from the non-hierarchical clustering of TEMI to a hierarchical arrangement of clusters. Their algorithm is not very complicated, but also it is not trivial. Because the authors do not compare against other ways of arranging TEMI clusters into a hierarchy, it is unclear whether their algorithm has any value.
> (2c) I do not think agglomerative clustering of image pixels is a strong baseline to compare against and agglomerative clustering of the embeddings would be a stronger baseline. It is important to know whether the method proposed beats this simple baseline.
>
> With regards to the inability to test on real-world hierarchical datasets, if the authors had provided more comprehensive baselines then there would be data points they could compare against. IMO this would strengthen the message of the paper, since it would emphasize how existing deep hierarchical methods scale poorly. I think the method proposed in the paper would also out-scale simply deploying agglomerative clustering on embeddings from a pretrained model (in terms of compute time), whilst out-performing it.

---

### Author Response · Authors · 2024-11-22
**Reply to all Reviewers**

Dear Reviewers,

Thank you all for the constructive and helpful feedback.

We appreciate the positive feedback regarding the effectiveness of our method (Reviewer abpg) and soundness of our experiments (Reviewer LaeG), as well as on the writing of our paper (Reviewers z6t7, AeLC, LaeG, k4QD) and the simplicity/elegance of our idea compared to alternative approaches (Reviewers z6t7, AeLC, LaeG, abpg). However, we also acknowledge the concerns expressed by the Reviewers. In summary in the rebuttal we address the following main concerns:

- **Reviewer z6t7** We further elaborate on our selection of datasets and baselines for the hierarchical clustering experiments. We justify our choices, highlighting the relevance of our showcased results within these experimental settings, that effectively validate the claims presented in the paper. To enhance clarity, we have revised Algorithm 1 in the updated manuscript, integrating the Reviewer's valuable suggestions.
- **Reviewer AeLC** We clarify that the efficient adaptation of a pre-trained flat clustering model to perform hierarchical clustering, as an alternative to designing ad-hoc complex approaches, is a key idea introduced in this work. We also further elaborate on our choices of baselines, and clarify how our run time comparison constitutes a fair comparison.
- **Reviewer LaeG** We clarify the key point that in Section 4.1 we consider a *fully unsupervised setup*, where our method achiveves remarkable performance. To complement the results in this section, we also include in the updated manuscript the hierarchical clustering performance obtained by our approach on the ImageNet1K dataset (Appendix C).
- **Reviewer k4QD** We remark that a key contribution of this work is introducing a different perspective on hierarchical clustering. Compared to existing works, we propose to efficiently adapt a pretrained flat model to infer a hierarchy of clusters, rather than focusing on designing costly and complex ad-hoc hierarchical approaches. We highlight that, while the idea of exploiting information in the logits to construct a hierarchy of classes was explored in classification, we are to the best of our knowledge the first ones to use this intuition for hierarchical clustering. Notably, we use this intuition to design a method for hierarchical clustering that overcomes the shortcomings in performance and scalability of current approaches.
- **Reviewer abpg** We provide useful clarifications, and elaboarate on the details of the aggregation function used in our experiments, for which we also include an ablation in the updated manuscript (Appendix C).

References for the whole rebuttal:
[1] Manduchi et al. Tree variational autoencoders. In Advances in Neural Information Processing Systems, 2023. [2] Mautz et al. DeepECT: The deep embedded cluster tree. Data Science and Engineering, 2020. [3] Chami et al. From trees to continuous embeddings and back: Hyperbolic hierarchical clustering. In Advances in Neural Information Processing Systems, 2020. [4] Wang et al. Internimage: Exploring large-scale vision foundation models with deformable convolutions. arXiv preprint arXiv:2211.05778, 2022.

---

### Meta-Review · Area_Chair_VhuM · 2024-12-21

**Metareview:**

This works critically examines deep learning approaches to hierarchical clustering, and proposes to instead infer a hierarchical clustering from pre-trained representations and flat clusterings. The main strengths are the simplicity and generality of the method, which can be applied to any input clustering that provides logits, the contrast of the method with existing divisive/top-down deep clustering methods, and improved results along with results on both unsupervised and supervised clustering (z6t7, AeLC, LaeG). The main weaknesses are the insufficient experiments (z6t7) and baselines (z6t7, AeLC, k4QD) and a minor weakness of showing results only for vision (AeLC) while clustering could apply to data of any modality. The submission is missing controlled comparisons between the proposed method and existing methods on the same and standard representations (z6t7, LaeG) and missing a baseline for the extension from flat to hierarchical clustering, such as agglomerative clustering (z6t7). The authors discuss these points, especially the choice of datasets and comparisons, but do not convince the reviewers (see the additional comments on the discussion).

Explanation of Decision:

Five reviewers with expertise on deep learning, clustering, meta-learning, and hierarchical modeling agree on rejection (z6t7: 3, LaeG: 3, k4QD: 3, AeLC: 5, abpg: 5). The meta-reviewer does not find cause to overrule the expert reviewers. On the balance, the weaknesses—especially the limited comparisons and datasets—outweigh the strengths of this work at this time. It is important to provide baselines and comparisons of overall performance, as done here, as well as controlled experiments that match representations/architectures/aggregation choices. Without controlled experiments, it is difficult to understand the significance of the proposed method and its results, so for this reason and in agreement with the overall evaluation by the reviewers the meta-reviewer sides with rejection. The authors are nevertheless encouraged to incorporate the precise feedback, for instance the suggestions from z6t7 on informative comparisons, and resubmit.

**Additional Comments On Reviewer Discussion:**

The authors reply to each review and provide a general response. 3/5 reviewers reply in turn: abpg increases their score to 5 given answers to their questions, AeLC maintains their score due to the lack of ablation and broader comparisons, and z6t7 maintains their score because additional baselines and real-world datasets were not considered and evaluated. The general response summarizes the main points made by the authors in reply to the reviews: justification of comparisons and datasets, the contribution of adapting flat clusters into hierarchical clusters, the distinction of fully unsupervised clustering and not, and further elaboration on the contributions to hierarchical clustering and the aggregation function used in this work. The reviewers who replied acknowledge these points, but find them insufficient and unsatisfactory on the whole. For the 2/5 reviewers who did not reply, the meta-reviewer closely examined the reviews and responses, and found the point about supervision addressed (LaeG), but did not find the points about different representations across baselines and the choice of "standard" models (like for instance ResNets or ViTs) to be resolved. For the avoidance of doubt, the decision incorporates all of the reviews and author responses, and has further weighted the complete threads with author-reviewer discussion.

While two authors rated the work as marginally below (rating = 5), and one raised their rating to this level, no reviewer championed the paper in the final discussion phase.

---

### Decision · Program_Chairs · 2025-01-22

Reject